# Variability Aware Recursive Neural Network (VARNN): A Residual-Memory Model for Capturing Temporal Deviation in Sequence Regression Modeling

## Abstract

Real-world time series data exhibit non-stationary behavior, regime shifts, and temporally varying noise (heteroscedastic) that degrade the robustness of standard regression models. We introduce the Variability-Aware Recursive Neural Network (VARNN), a novel residual-aware architecture for supervised time-series regression that learns an explicit error memory from recent prediction residuals and uses it to recalibrate subsequent predictions. VARNN augments a feed-forward predictor with a learned *error-memory* state that is updated from residuals over a short context steps as a signal of variability and drift, and then conditions the final prediction at the current time step. Across diverse dataset domains, appliance energy, healthcare, and environmental monitoring, experimental results demonstrate VARNN achieves superior performance and attains lower test MSE with minimal computational overhead over static, dynamic, and recurrent baselines. Our findings show that the VARNN model offers robust predictions under a drift and volatility environment, highlighting its potential as a promising framework for time-series learning.

Time-series regression, non-stationarity, residual learning, recurrent neural networks, distribution shift, system identification.

## 1 Introduction

Regression is one of the fundamental tasks in machine learning, where the goal is to learn a mapping from input features to a continuous-valued output. Classical regression models, such as linear regression, assume a static (stationary) relationship between predictors and target, whereas flexible learners (e.g., random forests and neural networks) capture nonlinear dependencies Hyndman & Athanasopoulos (2021). In many domains, however, the data are not independent and identically distributed (i.i.d.) but instead arrive as time series, where observations are temporally ordered and often correlated. Brockwell & Davis (2002).

This motivates time series regression, where the objective is to predict the output at each timestep using the current input and a short history of recent information (one-step-ahead foretasting, per-timestep supervised prediction) Hyndman & Athanasopoulos (2021); Brockwell & Davis (2002). In the machine learning literature, this is typically framed as linear or nonlinear regression with lagged features; in control systems and engineering, closely related formulations appear as Autoregressive with Exogenous Input (ARX) and Nonlinear ARX (NARX) models Ljung (1999); Lin et al. (1996). These perspectives are widely applied across energy analytics, biomedical monitoring, and environmental modeling, where local dynamics interact with exogenous drivers Candanedo et al. (2017); Pimentel et al. (2016); Zhang et al. (2017).

Beyond lagged-feature regression, time series can be framed as sequence modeling, where a model ingests a history of inputs (and optionally outputs) and produces either a single next-step prediction (sequence-to-one) or a trajectory (sequence-to-sequence). Recurrent neural networks (RNNs) and their variants (LSTM, GRU) implement this paradigm by maintaining a latent state that is updated recursively, enabling the model to summarize long contexts and capture nonlinear temporal dependencies Lipton et al. (2015).

While ARX/NARX and modern deep sequence models provide strong *dynamic* baselines, Ljung (1999); Lin et al. (1996); Lipton et al. (2015), real-world time series frequently exhibit variability, non-stationarity, and regime shifts driven by external disturbances, human behavior, or changing environmental conditions Hyndman & Athanasopoulos (2021); Brockwell & Davis (2002). Standard lagged-regression approaches model dependence on past outputs and inputs but do not explicitly represent how prediction errors themselves evolve. Deep architectures, such as Recurrent Neural Networks (RNNs), Long Short-Term Memory (LSTM) networks, and Gated Recurrent Units (GRUs) encodes temporal information in hidden states, yet typically capture variability only implicitly, treating residuals as a training signal rather than as a stateful source of information Lipton et al. (2015); Hochreiter & Schmidhuber (1997); Cho et al. (2014).

To address this limitation, we propose the Variability-Aware Recursive Neural Network (VARNN), a residual-aware neural regression framework [1]. Unlike ARX/NARX (or their ML equivalents), which use past outputs directly as regressors, VARNN introduces a residual memory mechanism that tracks and reuses prediction errors as an additional signal of variability. This enables the model to adapt to non-stationary conditions, offering greater robustness across diverse time series domains such as energy, biomedical, and environmental systems. Our main contributions are as follows:

1. We introduce the VARNN model, a novel residual-aware regression architecture that explicitly incorporates variability through error dynamics.
2. We systematically compare VARNN against three distinct families of strong regression baselines: static, dynamic lag and sequence models, including ARX, ARMAX, MLP, RNN, Dlinear and PatchTST.
3. We demonstrate through experiments on three distinct domain datasets that VARNN achieves superior robustness in the presence of noise, variability, and regime shifts.

## 2 RELATED WORK

**Static regression.** Classical regression methods, including linear regression, ridge, lasso, kernel models, and tree ensembles, are sample-efficient and interpretable but treat each $(\mathbf{x}_t, y_t)$ independently. They lack mechanisms to encode temporal dependence or adapt to time-varying uncertainty, leading to degraded accuracy under autocorrelation and non-stationarity (Hyndman & Athanasopoulos, 2021; De Gooijer & Hyndman, 2006).

**Dynamic regression.** Autoregressive models with exogenous inputs (ARX) and their nonlinear counterparts (NARX) incorporate lagged outputs and inputs, offering practical interpretability. However, they typically assume white, homoscedastic innovations and require careful lag-order selection, which limits robustness under distribution shift (Pankratz, 2012; Diversi et al., 2010; Clark et al., 2020). Neural extensions improve flexibility while preserving interpretability (Dong et al., 2025). The ARMAX approach adds moving average terms to capture serially correlated innovations, improving fit when shocks persist but still relying on fixed parametric assumptions and pre-specified error horizons.

**Sequence models.** RNNs, LSTMs, and GRUs learn temporal dependencies by maintaining latent states, but they absorb variability into hidden dynamics without distinguishing systematic structure from stochastic disturbances, leading to instability under non-stationarity (Hochreiter & Schmidhuber, 1997; Cho et al., 2014; Lipton et al., 2015; Lim & Zohren, 2021). Beyond RNNs, Temporal Convolutional Networks and Transformer-style forecasters extend receptive fields or apply attention mechanisms, with mixed robustness under distribution shift (Zeng et al., 2023; Bai et al., 2018; Lim et al., 2019; Zhou et al., 2021; Oreshkin et al., 2019).

**Variance and non-stationarity aware methods.** Recent approaches explicitly address non-stationarity. Hybrids integrate deep nets with volatility models (e.g., GARCH) (Han et al., 2024), normalization/denormalization (e.g., RevIN) improves robustness under distributional shift (Kim et al., 2022), and new architectures separate short-term fluctuations from long-term structure or weak-stationarize inputs (Baidya & Lee, 2024; Liu et al., 2025).

---

[1] The term "VARNN" in this work denotes a *Variability-Aware Recursive Neural Network*. The naming emphasizes the model's focus on variability and residual-memory dynamics. It is unrelated to the classical "Vector Autoregressive (VAR)" statistical model used in econometrics.

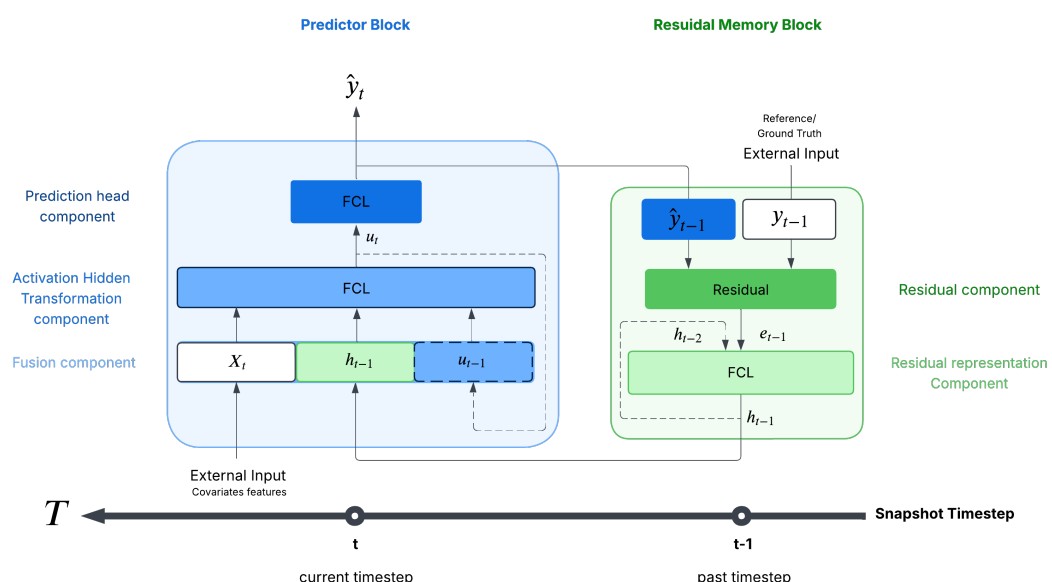

**Figure 1:** VARNN architecture: the Residual Memory Block (green) converts the innovation $e_\tau$ into a residual-memory state $\mathbf{h}_\tau$; the Predictor Block (blue) fuses $\mathbf{x}_\tau$ with $\mathbf{h}_{\tau-1}$ and, when enabled, $\mathbf{u}_{\tau-1}$, to produce $\hat{y}_\tau$.

**Positioning of VARNN.** Unlike hybrids that bolt variance models onto predictors or methods that normalize variability away, VARNN elevates recent innovations into a learnable residual-memory state and gates their influence on future predictions. This design makes variability actionable for adaptation while remaining lightweight and compatible with standard training pipelines.

## 3 METHODOLOGY

### 3.1 PROBLEM DEFINITION/SETUP

We consider supervised time–series regression with multivariate covariates and a univariate response:

$$\{(\mathbf{x}_\tau, y_\tau)\}_{\tau=1}^T, \qquad \mathbf{x}_\tau \in \mathbb{R}^d, \ y_\tau \in \mathbb{R}.$$

For each prediction time $t$, we construct a sliding window of length $w$:

$$\mathcal{W}_t := \{(\mathbf{x}_{t-w+1}, y_{t-w+1}), \ldots, (\mathbf{x}_{t-1}, y_{t-1}), \mathbf{x}_t\}, \tag{1}$$

where $y_t$ is unknown at prediction time. The goal is to predict the target $\hat{y}_t$ from the past $w-1$ labeled steps and the current covariates such that:

$$\hat{y}_t = f_\theta(\mathbf{x}_{t-w+1:t}, \ y_{t-w+1:t-1}). \tag{2}$$

The learning objective is to estimate the model parameters $\boldsymbol{\theta}$ by minimizing the mean squared error (MSE) over the $N$ training sampling windows:

$$\mathcal{L}(\boldsymbol{\theta}) = \frac{1}{N} \sum_{t=1}^N (y_t - \hat{y}_t)^2. \tag{3}$$

### 3.2 MODEL ARCHITECTURE

Variability-Aware Recursive Neural Network (VARNN) is a dynamic sequence regressor that explicitly encodes recent prediction errors into the next-step computation via a dedicated residual-memory pathway. Unlike classical RNNs, which update hidden states solely from inputs and prior activations, VARNN promotes the one-step innovation $e_\tau$ to a *residual-memory* state that calibrates the next prediction, making *prediction innovations* the primary recurrent signal. As shown in Figure 1, the architecture comprises two coupled blocks unrolled over the $w-1$ labeled context steps and used once more for the horizon prediction at time $t$:

**Predictor Block** The Predictor Block's purpose is to *map fused inputs to the latent representation* $\mathbf{u}_\tau$ *and produce the one-step prediction* $\hat{y}_\tau$. Given $\mathbf{x}_\tau \in \mathbb{R}^d$, residual memory state $\mathbf{h}_{\tau-1} \in \mathbb{R}^m$, we form the Fusion component as following:

$$\mathbf{z}_\tau = [\mathbf{x}_\tau; \mathbf{h}_{\tau-1}], \qquad \mathbf{z}_\tau \in \mathbb{R}^{d+m}. \tag{4}$$

We compute a latent representation by mapping the fused vector $\mathbf{z}_\tau$ to a $k$-dimensional activation through a nonlinear layer:

$$\mathbf{u}_\tau = \sigma(\mathbf{W}_z \mathbf{z}_\tau + \mathbf{b}_z), \qquad \mathbf{u}_\tau \in \mathbb{R}^k \tag{5}$$

where $\sigma(\cdot)$ is a pointwise nonlinearity (ReLU by default). This transformation is then projected to a linear scalar to estimate $\hat{y}_\tau$ as one–step prediction:

$$\hat{y}_\tau = \mathbf{W}_o \mathbf{u}_\tau + \mathbf{b}_o, \qquad \hat{y}_\tau \in \mathbb{R} \tag{6}$$

Concretely, at each supervised step $\tau$ ($\tau \leq t-1$), inputs $\mathbf{x}_\tau$ are fused with the previous residual memory $\mathbf{h}_{\tau-1}$, mapped to the latent representation, and read out to a prediction. It recursively implements the forward computation for $\hat{y}_\tau$ while refreshing the residual memory block.

**Residual Memory Block:** The Residual Memory block is an error-aware memory that encodes how prediction deviates from the current observation as a signal of variability and stores it as state that conditions the next step. The one-step prediction error is a reusable state that conditions the next step. Given the current supervised step prediction $\hat{y}_\tau$ and ground truth $y_\tau$, we form the innovation :

$$e_\tau = y_\tau - \hat{y}_\tau \in \mathbb{R}. \tag{7}$$

This innovation scalar $e_\tau$ is embedded into an $m$-dimensional residual-memory representation via a learnable projection and a pointwise nonlinearity activation function denoted as $\rho(\cdot)$. We update the memory state by:

$$\mathbf{h}_\tau = \rho(\mathbf{W}_\epsilon e_\tau + \mathbf{b}_\epsilon). \qquad \mathbf{h}_\tau \in \mathbb{R}^m. \tag{8}$$

By updating the state with the innovation, we treat the residual as a *primary recurrent input*. This *explicit error pathway* exposed variability and misspecification to the next-step predictor through the fusion term in (4). The *Predictor Block* performs the transformation of the input fusion and prediction; the *Residual-Memory Block* handles error assimilation. Together, they form a lightweight recurrence framework that emphasizes variability tracking.

**Current-time prediction** At current time step $t$, we predict $\hat{y}_t$ using the current covariates $x_t$ and last available memory state without performing a residual update:

$$\hat{y}_t = \mathbf{W}_o^\top \sigma(\mathbf{W}_z [\mathbf{x}_t; \mathbf{h}_{t-1}] + \mathbf{b}_z) + b_o. \tag{9}$$

Once $y_t$ is observed (offline or streaming), the residual memory can be refreshed via (8) for subsequent windows.

## 3.3 TRAINING OBJECTIVE AND OPTIMIZATION

We minimize the mean–squared error (MSE) on the current step prediction:

$$\mathcal{L}(\theta) = \frac{1}{N} \sum_{\text{windows}} (y_t - \hat{y}_t)^2, \quad \theta = \{\mathbf{W}_z, \mathbf{b}_z, \mathbf{W}_o, \mathbf{b}_o, \mathbf{W}_\epsilon, \mathbf{b}_\epsilon\}. \tag{10}$$

We use Adam with mini-batches of windows, early stopping on validation MSE, ReLU nonlinearities, and zero-state initialization: $\mathbf{h}_{t-w} = \mathbf{0}$.

## 3.4 MODEL VARIANTS

While the core VARNN architecture defines a residual-memory pathway that encodes innovations into a state $\mathbf{h}_\tau$, different update dynamics could affect how variability is represented and reused. Therefore, we study several update rules and augmentations that trade off responsiveness and stability. We instantiate four variants along two orthogonal design axes: (i) whether the residual memory update is *instantaneous* or *accumulative*, and (ii) whether the model also carries forward an *activation memory* $\mathbf{u}_{\tau-1}$ in addition to the residual state. This yields the four following configurations:

- **VARNN-RM** (*Residual Memory only*): This is the base model that uses the instantaneous update in (8), where the innovation $e_\tau$ is projected directly into the memory state. This lightweight design reacts only to the most recent error. This provides a sharp correction based solely on the most recent deviation.
- **VARNN-RM+AM** (*Residual + Activation Memory*): In addition to residual memory, the previous latent activation $\mathbf{u}_{\tau-1}$ is carried forward into the fusion vector in (4). The fused component of predictor input becomes $\mathbf{z}_\tau = [\mathbf{x}_\tau; \mathbf{h}_{\tau-1}; \mathbf{u}_{\tau-1}]$, allowing the predictor to integrate a compact nonlinear summary of recent fused inputs alongside the residual state.
- **VARNN-ARM** (*Accumulative Residual Memory*): which accumulates recent innovations into a smoother state. This variant behaves like a learned nonlinear autoregressive filter on prediction errors. In residual representation component (8), we replace the instantaneous update with an accumulative rule by:

$$\mathbf{h}_\tau = \rho(W_h \mathbf{h}_{\tau-1} + W_\epsilon e_\tau + b_\epsilon), \qquad \mathbf{h}_\tau \in \mathbb{R}^m. \tag{11}$$

which introduces persistence and enables the residual-memory state to integrate multiple past innovations. This design is motivated by settings with drift or systematic bias, where cumulative error provides a corrective signal.
- **VARNN-ARM+AM** (*Accumulative Residual + Activation Memory*): Combines accumulative residual memory with activation carry-over, yielding the richest variant. It retains nonlinear summaries of fused inputs while integrating error dynamics over time, aimed at regimes with both strong short-term variability and slow distributional shifts.

These four variants highlight different ways to leverage error information: RM emphasizes responsiveness, ARM emphasizes persistence, AM augments representational capacity, and their combinations balance the two. We empirically compare all variants in Section 5 to assess which design choices best enhance robustness under non-stationary conditions.

### 3.5 COMPUTATIONAL COMPLEXITY

Let $d$ be the covariate dimension, $k$ the predictor width, and $m$ the residual–memory width (default $m=d$). Relative to a covariate-only feed-forward predictor with fusion weight $W_z \in \mathbb{R}^{k \times d}$, the base model **VARNN-RM** concatenates $\mathbf{h}_{\tau-1} \in \mathbb{R}^m$, expanding $W_z$ to $\mathbb{R}^{k \times (d+m)}$ and adding $km$ parameters and $km$ multiplies per step. The residual projection $W_r \in \mathbb{R}^{m \times 1}$ adds $m$ parameters and $m$ multiplies. Ignoring biases, the incremental overhead is therefore $O(km)$ parameters and multiplies. By contrast, a standard RNN update $\mathbf{h}_\tau = \rho(W_x \mathbf{x}_\tau + W_h \mathbf{h}_{\tau-1})$ costs $O(dk + k^2)$ per step. Since $m \ll k$ in practice, the residual pathway adds negligible overhead relative to an RNN cell. *Note: the linear readout head is shared across models and is omitted from these counts.*

### 3.6 ALGORITHMIC VIEW

Algorithm 1 summarizes the teacher-forced unroll over the $(w-1)$ context steps and the current-time prediction for the base model **VARNN-RM**. For all other variants, see subsection E.1

---

**Algorithm 1** VARNN: Windowed prediction at time $t$

---

**Require:** Window $\mathcal{W}_t = \{(\mathbf{x}_{t-w+1}, y_{t-w+1}), \ldots, (\mathbf{x}_{t-1}, y_{t-1}), \mathbf{x}_t\}$; parameters $\theta = \{\mathbf{W}_z, \mathbf{b}_z, \mathbf{W}_o, \mathbf{b}_o, \mathbf{W}_r, \mathbf{b}_r\}$
**Ensure:** Current prediction $\hat{y}_t$
 1: **Init residual memory:** $\mathbf{h}_{t-w} \leftarrow \mathbf{0}$
 2: **for** $\tau = t-w+1$ **to** $t-1$ **do**                    ▷ teacher-forced context (supervised) steps
 3:     $\mathbf{z}_\tau \leftarrow [\mathbf{x}_\tau; \mathbf{h}_{\tau-1}]$
 4:     **Inference:** $\mathbf{u}_\tau \leftarrow \theta(\mathbf{W}_z \mathbf{z}_\tau + \mathbf{b}_z)$;   $\hat{y}_\tau \leftarrow \mathbf{W}_o \mathbf{u}_\tau + \mathbf{b}_o$
 5:     **Innovation:** $e_\tau \leftarrow y_\tau - \hat{y}_\tau$
 6:     **Residual update:** $\mathbf{h}_\tau \leftarrow \rho(\mathbf{W}_r e_\tau + \mathbf{b}_r)$
 7:                                       ▷ current-time prediction (no residual update at $t$)
 8: $\mathbf{z}_t \leftarrow [\mathbf{x}_t; \mathbf{h}_{t-1}]$
 9: $\hat{y}_t \leftarrow \mathbf{W}_o \theta(\mathbf{W}_z \mathbf{z}_t + \mathbf{b}_z) + \mathbf{b}_o$
10: **return** $\hat{y}_t$

---

**Notes.** The base model **VARNN-RM** uses an *instantaneous residual-memory update*: $\mathbf{h}_\tau \leftarrow \rho(\mathbf{W}_r e_\tau + \mathbf{b}_r)$, where $\rho(\cdot)$ can be ReLU or tanh depending on the stability requirements. This design makes innovations the sole recurrence signal carried forward across steps.

### 3.7 VARNN vs Standard recurrent models (RNN/LSTM/GRU)

Conventional recurrent models update a hidden state from $(\mathbf{x}_\tau, \mathbf{h}_{\tau-1})$ and learn to accommodate variability implicitly through gradients. VARNN makes the *innovation process* a first-class citizen by dedicating a pathway (RM/ARM) that injects prediction error back into the state, explicitly targeting drift, heteroscedasticity, and short-horizon miscalibration.

## 4 Experiment Settings

We evaluate our proposed model **VARNN** on multivariate sequence-to-one regression (also called one step ahead forecasting) across three domains (energy, healthcare, environment). The goal is to test whether explicit residual-memory yields robustness under non-stationarity relative to srong static, dynamic (lagged), and sequence baselines, under identical preprocessing, splits, and training protocols.

**Datasets.** We use nine popular datasets grouped by domain summarized in the Table 1. All tasks are scalar regression per time step as following:

- **Energy.** (i) **Appliances Energy Prediction** (*Appliances*) records household power consumption with indoor/outdoor weather and occupancy covariates; the target is appliance energy use (Wh) (Candanedo et al., 2017). (ii) **ETTh1** and (iii) **ETTh2** are the electricity transformer temperature benchmarks with hourly electricity load and meteorological features to predict the oil temperature from exogenous covariates (Zhou et al., 2021).
- **Healthcare.** We use three scalar targets derived from the BIDMC PPG/ECG recordings (Pimentel et al., 2016): (iv) **BIDMC HR** (heart rate), (v) **BIDMC RR** (respiratory rate), and (vi) **BIDMC SPO2** (oxygen saturation). In all cases, the covariates are three channels short-term waveform statistics and auxiliary signals (PPG, RESP, ECG) at 125 Hz recorded from 52 patients.
- **Environmental.** (vii) **Beijing PM2.5** and (viii) **Beijing PM10** contain air-quality and meteorological variables over 12 monitoring stations; we predict $PM_{2.5}$ and $PM_{10}$ concentrations respectively (Zhang et al., 2017). (ix) **WEATHER** is a multi-variate meteorological time series where the target is the weather temperature single scalar (Zhou et al., 2021).

**Table 1:** Dataset summary (post-cleaning): size, missingness, feature count $d$, and scalar target.

| Dataset | Size | Missing | #Feat. ($d$) | Target |
|---|---|---|---|---|
| **Energy** | | | | |
| Appliances | 19,735 | No | 27 | Appliance energy use (Wh) |
| ETTh1 | 17,420 | No | 6 | Oil temperature |
| ETTh2 | 17,420 | No | 6 | Oil temperature |
| **Healthcare (BIDMC)** | | | | |
| BIDMC–HR | 25,436 | No | 375 | Heart rate (bpm) |
| BIDMC–RR | 25,436 | No | 375 | Respiratory rate (breaths/min) |
| BIDMC–SPO2 | 25,436 | No | 375 | Oxygen saturation ($SpO_2$) |
| **Environmental** | | | | |
| Beijing–PM2.5 | 420,768 | Yes | 9 | $PM_{2.5}$ concentration |
| Beijing–PM10 | 420,768 | Yes | 9 | $PM_{10}$ concentration |
| WEATHER | 35,064 | No | 6 | Temperature (°C) |

**Preprocessing, windowing, and split.** Continuous inputs and targets are min–max scaled using training statistics and applied unchanged to validation/test. We adopt sequence-to-one with window length $w=5$ and stride 1: the first $w-1$ steps warm up residual memory; the model predicts the final step. Splits are chronological (80/20 train/test). Windows never cross split boundaries. For the PM2.5 and PM10 datasets, we filled missing values per station before scaling. Both are subject-wise splits (per station). BIDMC HR, RR, and SPO2 consist of three channels (PPG, RESP, ECG) at 125 Hz; we partition the signal into 1-second bins and concatenate the 125 samples per channel (no overlap), yielding 375 input vector features per time step. Each dataset is a subject-wise split (per patient).

**VARNN Variants.** We report two primary variants in the main table: **VARNN–RM**: instantaneous residual-memory update and **VARNN–RM+AM**: residual-memory + activation-memory fusion. The

Two accumulative variants (ARM, ARM+AM) are included in the Appendix; they exhibit comparable accuracy with improved stability in highly variable environments. We report the two primary variants (RM and RM+AM) in Table 2. The two accumulative variants (ARM, ARM+AM) are included in the Appendix achieve comparable accuracy with improved stability, as detailed in Appendix D.1.

**Baselines.** We compare VARNN against thirteen baselines grouped into three broad regression families: static, dynamic lag, and dynamic sequence regression, all trained under identical preprocessing, scaling, and chronological splits. **(1) Static Regressors:** these models use only contemporaneous covariates $\mathbf{x}_t$ with no temporal history. They include Linear Regression (LR), Random Forest (RF), and a Multilayer Perceptron (MLP), each learning a direct mapping $f(\mathbf{x}_t) \rightarrow y_t$. **(2) Dynamic Lag Regressors:** These models incorporate explicit lag concatenation over the same horizon used by VARNN windowing ($w{=}5$). ARx-LR uses only lagged targets and current exogenous covariates $\mathbf{x}_t$, ARX-LR incorporates lagged covariates and lagged targets with current $\mathbf{x}_t$, NARX-RF and NARX-MLP provide nonlinear autoregressive variants; and ARMAX-LR extends ARX with moving–average innovation terms. DLinear (Zeng et al., 2023) applies trend–seasonal decomposition with a linear head over lagged covariates and $\mathbf{x}_t$. **(3) Dynamic Sequence Regressors:** These models treat the windowed covariates $\mathbf{x}_{t-w+1:t}$ as a temporal sequence. We evaluate RNN, LSTM, and GRU, each predicting via a linear head on the final hidden state. We also evaluate PatchTST the SOTA transformer-based sequence baseline using a patching mechanism Nie et al. (2023).

**Hyperparameters, training, and libraries.** Neural models (VARNN, RNN/LSTM/GRU, and MLP) use hidden size 128, ReLU activations, and are implemented in TensorFlow. SOTA models DLinear and PatchTST are implemented using the official PyTorch code from their original repositories; PatchTST is configured with embedding dimension $d_{\text{model}}{=}128$, feedforward dimension 128, 4 attention heads, one Transformer encoder layers, patch length 3, stride 1. ALL neural models are trained with a linear a one step regression head with Adam (lr $= 3{\times}10^{-3}$), batch size 128, for up to 50 epochs with early stopping on validation MSE (patience 50 with best–model restoration). Random forest models use 500 trees with scikit–learn defaults. Linear regression baselines (LR, ARx–LR, ARX–LR) also use scikit–learn. The ARMAX is implemented via statsmodels library, with autoregressive and moving average orders chosen p and q=4 to match the same lag window and exogenous structure used by the other dynamic regression baselines.

**Evaluation Protocol.** We evaluated and reported the models based on the mean squared error (MSE) at the epoch with the best validation MSE. Data splits and parameter initializations are fixed under the same seed number (seed=2025).

$$\text{MSE} \;=\; \frac{1}{N}\sum_{i=1}^{N}\left(y_w^{(i)} - \hat{y}_w^{(i)}\right)^2.$$

## 5 Results

We evaluate models under the protocol described in Sec. 4 and report mean squared error (MSE; lower is better) on the training and test splits of all three datasets. Models are grouped into *Static* baselines (no lag) and *Dynamic* regressors (explicit lags or sequence). Table 2 summarizes the results.

**Overall performance.** VARNN consistently achieves the lowest test MSE across all datasets. On *Appliances*, the strongest baseline ARX-LR reaches $5.04 \times 10^{-3}$, while VARNN-RM base model improves to $3.28 \times 10^{-3}$, marking a $\sim 35\%$ reduction. On *BIDMC HR*, ARX-LR yields $2.7 \times 10^{-4}$ test error, but VARNN-RM reduces this to $1.5 \times 10^{-4}$, almost halving the error $\sim 44.4\%$ . On *Beijing PM2.5*, VARNN-RM reaches $2.6 \times 10^{-4}$ compared to $4.9 \times 10^{-4}$ for ARX-LR, again nearly a twofold improvement $\sim 47\%$. In all cases, the gap between training and test MSE remains small, underscoring robustness and resistance to overfitting.

**Static vs. Dynamic** Static regressors that ignore temporal structure substantially underperform dynamic families on every dataset. RandomForest, in particular, shows pronounced overfitting: near-zero training error but much larger test error. This highlights the need for temporal context.

**Lags vs. Sequence models.** Explicit-lag baselines (ARX, NARX, ARMAX) generally improve over static models but can be sensitive to lag design and model capacity. Classical recurrent baselines (RNN, LSTM, GRU) deliver mixed results: competitive on *Appliances* and *PM2.5*, but unstable in other tasks. In contrast, VARNN attains both strong accuracy and stable generalization across domains, confirming the benefit of residual memory recurrence under non-stationarity.

**Table 2:** Train and test mean squared error (MSE; ↓) on all datasets. *Static* models use only contemporaneous covariates $\mathbf{x}_t$; *Dynamic* models additionally incorporate lagged outputs/inputs or recurrent states. Best test scores are in **bold**.

| Model | ENERGY | | | | | | Healthcare | | | | | | Environmental | | | | | |
| --- | --- | --- | --- | --- | --- | --- | --- | --- | --- | --- | --- | --- | --- | --- | --- | --- | --- | --- |
| | Appliances | | ETTh1 | | ETTh2 | | BIDMC HR | | BIDMC RR | | BIDMC SPO2 | | Beijing PM2.5 | | Beijing PM10 | | WEATHER | |
| | Train | Test | Train | Test | Train | Test | Train | Test | Train | Test | Train | Test | Train | Test | Train | Test | Train | Test |
| *Static (no lags)* | | | | | | | | | | | | | | | | | | |
| LR | 0.00799 | 0.00657 | 0.02792 | 0.03335 | 0.02243 | 0.04112 | 0.01830 | 0.02256 | 0.00984 | 0.00822 | 0.03659 | 0.03103 | 0.00212 | 0.00171 | 0.00375 | 0.00242 | 0.01516 | 0.01483 |
| RF | 0.00057 | 0.04815 | 0.00234 | 0.02888 | 0.00174 | 0.03274 | 0.00031 | 0.01842 | 0.00032 | 0.00705 | 0.00057 | 0.03925 | 0.00012 | 0.00162 | 0.00025 | 0.00227 | 0.00229 | 0.01235 |
| MLP | 0.00833 | 0.00778 | 0.00833 | 0.00778 | 0.02622 | 0.02049 | 0.00216 | 0.00168 | 0.01276 | 0.00933 | 0.04683 | 0.03452 | 0.00216 | 0.00168 | 0.00392 | 0.00224 | 0.01413 | 1.21301 |
| *Dynamic (lags)* | | | | | | | | | | | | | | | | | | |
| Dlinear | 0.01317 | 0.01076 | 0.54476 | 0.30845 | 0.45580 | 0.43386 | 0.04563 | 0.10959 | 0.04307 | 0.08024 | 0.10516 | 0.12605 | 0.01346 | 0.01338 | 0.01477 | 0.01468 | 0.39243 | 0.40353 |
| ARx–LR | 0.00395 | 0.00958 | 0.00041 | 0.04819 | 0.00030 | 0.24168 | 0.00069 | 0.03362 | 0.00041 | 0.01834 | 0.00100 | 0.05868 | 0.00028 | 0.00274 | 0.00087 | 0.00290 | 0.00042 | 0.34684 |
| ARX–LR | 0.00610 | 0.00504 | 0.00067 | 0.00033 | 0.00076 | 0.00075 | 0.00023 | 0.00027 | 0.00030 | 0.00052 | 0.00028 | 0.00046 | 0.00051 | 0.00049 | 0.00144 | 0.00091 | 0.00061 | 0.00069 |
| NARX–RF | 0.00079 | 0.02750 | 0.00008 | 0.00036 | 0.00008 | 0.00085 | 0.00003 | 0.00035 | 0.00004 | 0.00053 | 0.00004 | 0.00068 | 0.00005 | 0.00049 | 0.00015 | 0.00093 | 0.00006 | 0.00051 |
| NARX–MLP | 0.00652 | 0.00527 | 0.00074 | 0.00036 | 0.00082 | 0.00089 | 0.00065 | 0.00079 | 0.01390 | 0.01381 | 0.00065 | 0.00098 | 0.00053 | 0.00052 | 0.00147 | 0.00094 | 0.00079 | 0.00108 |
| ARMAX-LR | 0.00411 | 0.01002 | 0.00038 | 0.02952 | 0.00027 | 0.14067 | 0.00086 | 0.03092 | 0.00050 | 0.01136 | 0.00132 | 0.04648 | 0.00027 | 0.06462 | 0.00084 | 0.00429 | 0.00035 | 0.29126 |
| *Dynamic (Sequence)* | | | | | | | | | | | | | | | | | | |
| PatchTST | 0.00604 | 0.00689 | 0.01560 | 0.03251 | 0.03077 | 0.04006 | 0.01961 | 0.01900 | 0.00921 | 0.00878 | 0.05707 | 0.04502 | 0.00450 | 0.00492 | 0.00673 | 0.00621 | 0.04191 | 0.03640 |
| RNN | 0.00765 | 0.00697 | 0.01618 | 0.01779 | 0.01608 | 0.02246 | 0.00221 | 0.01263 | 0.00220 | 0.00811 | 0.00780 | 0.04480 | 0.00140 | 0.00150 | 0.00257 | 0.00209 | 0.00892 | 0.00973 |
| LSTM | 0.00813 | 0.00652 | 0.01414 | 0.01762 | 0.01406 | 0.01997 | 0.00125 | 0.00930 | 0.00201 | 0.00746 | 0.03392 | 0.02678 | 0.00097 | 0.00143 | 0.00257 | 0.00210 | 0.00846 | 0.00928 |
| GRU | 0.00824 | 0.00654 | 0.01352 | 0.02116 | 0.01576 | 0.02170 | 0.00145 | 0.00825 | 0.00274 | 0.00729 | 0.03381 | 0.02802 | 0.00110 | 0.00145 | 0.00200 | 0.00202 | 0.00834 | 0.00908 |
| **VARNN–RM** | **0.00412** | **0.00328** | **0.00037** | **0.00020** | **0.00041** | **0.00048** | **0.00016** | **0.00015** | **0.00017** | **0.00031** | **0.00015** | **0.00024** | **0.00024** | **0.00026** | **0.00085** | **0.00058** | **0.00028** | **0.00045** |
| **VARNN–RM+AM** | **0.00378** | **0.00329** | **0.00035** | **0.00018** | **0.00044** | **0.00049** | **0.00016** | **0.00015** | **0.00017** | **0.00029** | **0.00015** | **0.00022** | **0.00022** | **0.00025** | **0.00077** | **0.00057** | **0.00037** | **0.00042** |

**VARNN variants: RM vs RM+AM** The learning curve in Figure 2 shows that the RM+AM variant reveals a faster convergence speed advantage over the base model RM with broadly similar or better generalization. This finding suggests that adding the activation memory (AM), which carries the previous latent activation, enriches short-term temporal dynamics and stabilizes predictions under drift and noise.

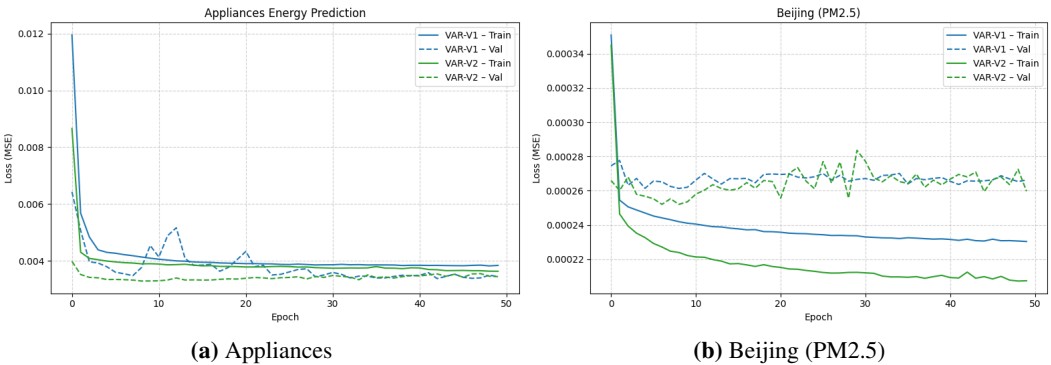

(a) Appliances         (b) Beijing (PM2.5)

**Figure 2:** Learning curves for VARNN-RM and VARNN-RM+AM across Appliances and Beijing datasets.

## 5.1 ABLATIONS

We conduct targeted ablations on the residual-memory mechanism in the base model **VARNN-RM**. All ablations use the same data splits, window length $W$, feature dimension $d$, activation hidden units $k$ and residual memory width $m$, optimizer (Adam, lr $= 10^{-3}$), batch size (128), early stopping on validation MSE with best-weights restore, and a fixed training budget (50 epochs max), unless otherwise stated. Each configuration is run over the same seed (2025) and reported on MSE loss.

**Effect of Residual Memory** Figure 3 compares training and validation learning curves for three variants: (i) *no residual* (the predictor receives a zero residual state), (ii) *residual memory (RM)*, and (iii) *accumulative residual memory (ARM)*. Across both PM2.5 and Appliances, residual-aware models converge more quickly and attain substantially lower test MSE than the baseline. RM alone already delivers most of the improvement, while ARM yields marginal additional gains and stability. The results confirm that feeding back prediction errors as a learnable memory state improves stability and generalization compared to models that absorb variability only implicitly.

**Residual-memory: scalar vs. projected** ($\mathbf{h}_\tau \in \mathbb{R}^m$). We compare two residual pathways in **VARNN-RM**: (i) a *scalar* state that carries the innovation $e_\tau$ as a single unit, and (ii) a *projected* memory that maps the innovation into an $m$-dimensional vector $\mathbf{h}_\tau = \rho(W_\epsilon e_\tau + \mathbf{b}_\epsilon) \in \mathbb{R}^m$. All other

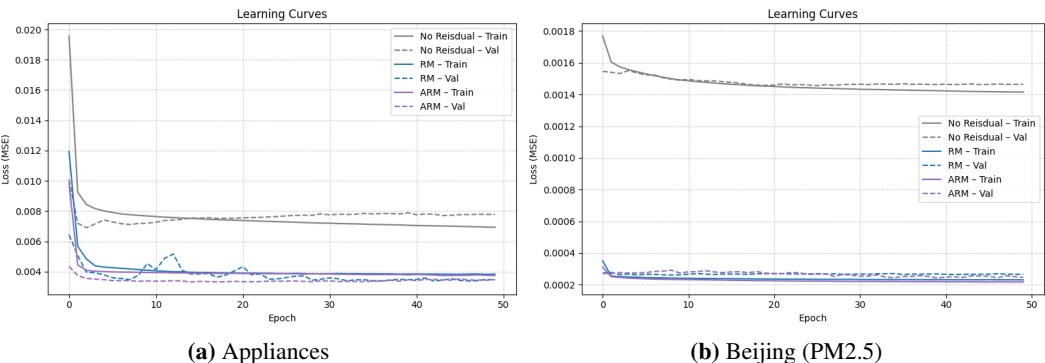

**(a)** Appliances             **(b)** Beijing (PM2.5)

**Figure 3:** Learning curves for VARNN-RM and VARNN-RM+AM across Appliances and Beijing datasets.

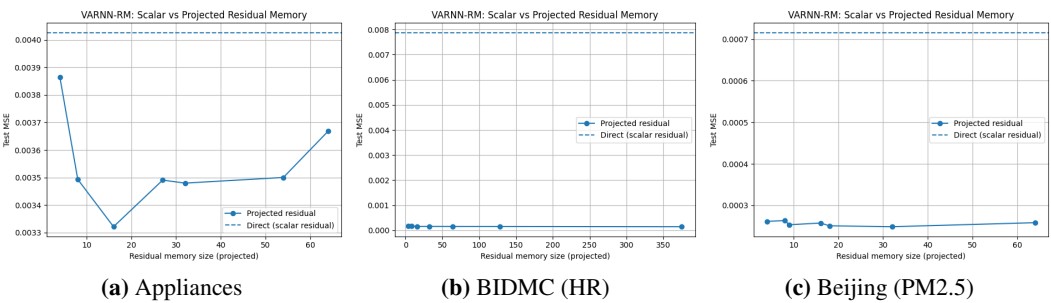

**(a)** Appliances        **(b)** BIDMC (HR)        **(c)** Beijing (PM2.5)

**Figure 4:** Residual memory size vs scalar across three datasets.

settings are fixed. We sweep $m \in \{4, 8, 16, 32, 64, \min(128, 2d), d\}$ and report test MSE ($\downarrow$). As shown in Figure 4, projecting the residual consistently outperforms the scalar variant: *Appliances:* scalar 0.004026 vs. best projected 0.003321 ($m{=}16$), **17.5%** lower; *BIDMC HR:* scalar 0.007978 vs. best 0.000154 ($m{=}64$), **98.1%** lower; *Beijing PM2.5:* scalar 0.000702 vs. best 0.000246 ($m{=}4$), **65.0%** lower. These results indicate that a single value cannot represent diverse error regimes (sign asymmetry, magnitude bands, burstiness) and may cause noise, thereby downgrading the model's performance. In contrast, a learned vector memory provides a richer residual representation and yields lower error. The optimal $m$ is dataset dependent: small on PM2.5, moderate on Appliances, and larger on BIDMC due to richer waveform dynamics.

# 6 ETHICS STATEMENT

This work develops supervised learning methods for time-series regression. Ethical risks are low and domain-dependent. When applied to sensitive domains (e.g., healthcare), ensure appropriate consent, privacy safeguards, and fairness audits.

# 7 CONCLUSION AND FUTURE WORK

We introduced the Variability-Aware Recursive (VARNN) network, a recurrent architecture that elevates prediction residuals to first-class signals via an explicit error memory state. Unlike standard lagged regression or conventional RNNs, VARNN updates a dedicated memory from past deviations and conditions subsequent predictions on this variability summary. Across three domains (*Energy*, *Healthcare*, and *Environmental*), VARNN delivered the lowest test MSE over all baselines, reducing MSE loss by roughly 35–50% versus strong ARX/NARX baselines and by factors of 2–6 versus RNNs, with $>80\%$ reductions over static models and up to $\sim$98% on *BIDMC HR*. Our model exhibits the potential to be the basis framework for future work of Time series regression. In future work, we plan to apply VARNN model to new domains characterized by volatile, regime-shifting dynamics such *I/O write-time prediction in HPC systems*, where contention, job mix, and filesystem effects induce bursty nonstationarity. Finally, we will extend VARNN *beyond one-step point regression* to multi-step-ahead forecasting.

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

## A  DISCLOSURE OF AI ASSISTANCE

All conceptual development, modeling, experiments, and analysis were carried out by the authors. Portions of the writing were refined using large language models (LLMs), which supported grammar, clarity, and formatting.

## B  REPRESENTATION LEARNING OF VARIABILITY IN VARNN

VARNN learns a representation of variability by encoding each one step prediction innovation $e_\tau = y_\tau - \hat{y}_\tau$ into an multi-dimensional residual memory vector $\mathbf{h}_\tau \in \mathbb{R}^m$ (Eq. 8, 11). Rather than passing the innovation as a raw scalar as used approach in the ARMAX model or error injection heuristics models, VARNN projects the residual through a learnable transformation that captures richer structure:

$$\mathbf{h}_\tau = \rho(\mathbf{W}_\epsilon e_\tau + \mathbf{b}_\epsilon) \qquad \text{(Residual Memory (RM) variant)}$$

In the accumulative variant (ARM), this embedding composes recursively with the previous memory state, forming a learned autoregressive filter on recent deviations:

$$\mathbf{h}_\tau = \rho(W_h \mathbf{h}_{\tau-1} + W_\epsilon e_\tau + b_\epsilon) \qquad \text{(Accumulative Residual Memory (ARM) variant)}$$

This projection allows the residual memory to encode richer properties of variability that a scalar cannot express such as magnitude, sign asymmetry, heteroscedastic volatility, burstiness, and short-term bias or drift regimes. The resulting memory state is then fed back into the predictive pathway via the fusion vector $\mathbf{z}_\tau$ (Eq. 4), enabling each next prediction to be conditioned on a learned summary of recent deviations.

In contrast, models that directly ingest the raw error as an additional input feature have no mechanism to shape or filter the noise structure of innovations, which empirically leads to noise amplification and performance degradation. For example, under identical training protocol and windowing size w=5 , ARMAX–LR exhibits substantially worse accuracy across datasets, most notably on ETTh1 (**0.02952** vs. VARNN–RM's **0.00020**) and ETTh2 (**0.14067** vs. VARNN–RM's **.00048**), showing that scalar innovation injection amplifies noise rather than providing a stable conditioning signal (see Results, Table 2). Similar degradation appears in healthcare and environmental datasets, where ARMAX–LR consistently ranks among the weakest performers.

As shown in our ablation study (§5.1), using a scalar residual state yields substantially worse test MSE across all three domains for example, replacing the scalar with a projected memory vector reduces error by $17.5\%$ on Appliances, $65\%$ on PM2.5 and $98\%$ on BIDMC HR. Thus, VARNN does not simply reuse prediction errors, it learns a structured residual representation that filters noise, stabilizes optimization, and preserves meaningful variability patterns for next-step prediction. This makes VARNN fundamentally distinguishing from ARMAX, ARX-style or recurrent models that absorb variability only implicitly, providing a principled and empirically validated approach to handling non-stationarity.

## C  ADDITIONAL IMPLEMENTATION AND ANALYSIS

The main paper establishes that residual memory improves robustness under non-stationarity. This appendix provides complementary implementation details and extended analyses to support the main results presented in Section 5. We first investigate the effect of the activation function within the residual-memory block (§D), comparing RELU and TANH activations to understand their impact on convergence stability and noise sensitivity. We then analyze the behavior of the four VARNN variants: RM, RM+AM, ARM, and ARM+AM on the Beijing PM2.5 dataset (§D.1), illustrating how residual accumulation and activation carry jointly affect temporal adaptation under non-stationary dynamics. Finally, we provide algorithmic and quantitative summaries of all configurations (§E.1, §E.2), including detailed pseudocode and per-variant performance metrics across datasets. Together, these results offer a deeper understanding of VARNN's internal mechanisms, training behavior, and architectural trade-offs, reinforcing its design rationale and empirical consistency across diverse temporal regimes.

## D ACTIVATION FUNCTION IN RESIDUAL-MEMORY BLOCK (RELU VS. TANH)

We conducted an ablation to examine the effect of the residual-memory activation $\rho(\cdot)$ in VARNN-RM, comparing the default RELU against a bounded TANH. Results are reported on two representative datasets: *Appliances Energy Prediction* and *Beijing PM2.5*. The learning curves graphs in Figure 5 show the train/validation learning curves over 50 epochs.

On **Appliances**, both activations converge to a similar validation floor ($\approx 3.9 \times 10^{-3}$ MSE), but TANH exhibits smoother trajectories with reduced oscillations near convergence. In contrast, RELU shows higher variance across epochs, suggesting sensitivity to local error bursts.

On **PM2.5**, the difference is negligible: both activations stabilize around $2.6 \times 10^{-4}$ MSE, with TANH again displaying slightly lower validation variance. The bounded range of TANH likely dampens extreme residual updates, which can be beneficial under noisy or volatile conditions.

Overall, these results indicate that while RELU and TANH achieve comparable final accuracy, TANH provides smoother convergence and more stable generalization in non-stationary settings. Unless sparsity in the residual embedding is explicitly desired, we recommend TANH as a default activation in the residual-memory block.

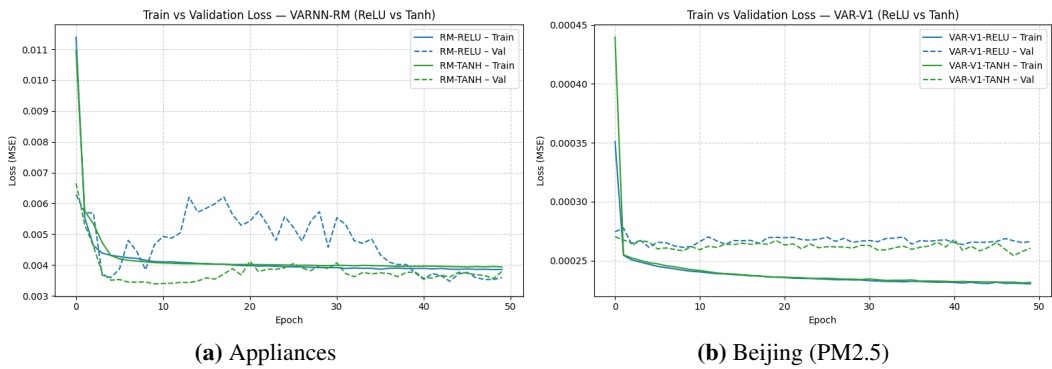

**(a)** Appliances            **(b)** Beijing (PM2.5)

**Figure 5:** Train vs. validation loss on *Beijing PM2.5* using VARNN-RM with ReLU vs. Tanh activations.

### D.1 VARIANTS PERFORMANCE ON BEIJING PM2.5

Figure 6 compares training and validation learning curves across the four VARNN variants: *residual memory (RM)*, *residual+activation memory (RM+AM)*, *accumulative residual memory (ARM)*, and *ARM+AM*:

**Residual memory (RM).** The base model VARNN-RM converges steadily but exhibits the highest validation loss among variants, with noticeable fluctuations during the first 20 epochs. This indicates that a pure residual pathway provides stability but underfits the variability in PM2.5 dynamics.

**RM+AM.** Adding activation memory improves training convergence but validation loss remains noisy, suggesting that activation carry alone does not robustly capture nonstationary fluctuations in air-quality data.

**ARM.** ARM achieves Lowest validation loss with smooth convergence after $\sim$10–15 epochs, demonstrating that the accumulating residuals are crucial on PM2.5. This demonstrates the benefit of accumulating residuals over time to encode volatility regimes.

**ARM+AM.** Training loss is the lowest, and validation performance closely tracks ARM, typically marginally higher or delayed by a few epochs.

In summary, on Beijing PM2.5, **ARM** achieves the best validation stability, with **ARM+AM** performing comparably. Therefore, the key driver is residual *accumulation*; activation offers little additional benefit for this dataset.

## E EXTENDED RESULTS AND FIGURES

### E.1 ALGORITHMIC VIEW OVER ALL VARNN VARIANTS

Algorithm 2 summarizes the teacher-forced unroll over the $(w-1)$ labeled context steps and the current-time prediction at $t$. We use $\theta(\cdot)$ for the predictor nonlinearity (e.g., ReLU) and $\rho(\cdot)$ for the residual-memory update (ReLU or $\tanh$). The four configurations correspond to: **RM** (residual

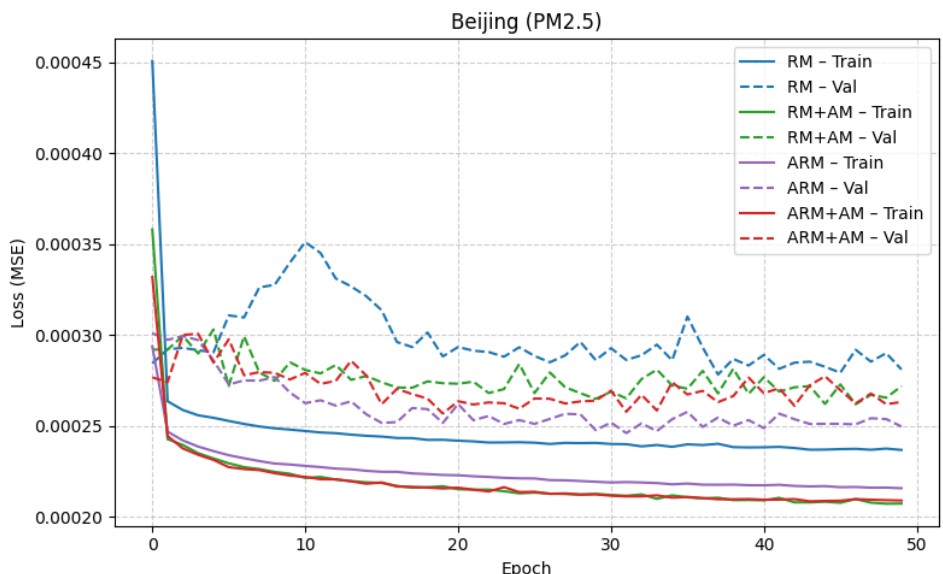

**Figure 6:** Beijing PM2.5: learning curves for VARNN variants. Accumulating residuals (ARM) improves stability and validation accuracy, while combining accumulation with activation memory (ARM+AM) yields similar performance with slightly smoother training dynamics.

memory only), **RM+AM** (residual + activation memory), **ARM** (accumulative residual memory), and **ARM+AM** (accumulative residual + activation memory).

---

**Algorithm 2** VARNN (RM, RM+AM, ARM, ARM+AM): Windowed prediction at time $t$

---

**Require:** Window $\mathcal{W}_t = \{(\mathbf{x}_{t-w+1}, y_{t-w+1}), \ldots, (\mathbf{x}_{t-1}, y_{t-1}), \mathbf{x}_t\}$; configuration $\in$ {RM, RM+AM, ARM, ARM+AM}; parameters $\theta = \{\mathbf{W}_z, \mathbf{b}_z, \mathbf{W}_o, \mathbf{b}_o, \mathbf{W}_r, \mathbf{W}_h, \mathbf{b}_r\}$

**Ensure:** Current prediction $\hat{y}_t$

1: Initialize residual memory $\mathbf{h}_{t-w} \leftarrow \mathbf{0}$
2: **if** configuration $\in$ {RM+AM, ARM+AM} **then**
3:      Initialize activation memory $\mathbf{u}_{t-w} \leftarrow \mathbf{0}$
4: **for** $\tau = t - w + 1$ **to** $t - 1$ **do**                      ▷ teacher-forced context steps
5:      **if** configuration $\in$ {RM, ARM} **then**
6:          $\mathbf{z}_\tau \leftarrow [\mathbf{x}_\tau; \mathbf{h}_{\tau-1}]$
7:      **else**
8:          $\mathbf{z}_\tau \leftarrow [\mathbf{x}_\tau; \mathbf{h}_{\tau-1}; \mathbf{u}_{\tau-1}]$
9:      $\mathbf{u}_\tau \leftarrow \theta(\mathbf{W}_z \mathbf{z}_\tau + \mathbf{b}_z)$
10:     $\hat{y}_\tau \leftarrow \mathbf{W}_o \mathbf{u}_\tau + \mathbf{b}_o$
11:     $e_\tau \leftarrow y_\tau - \hat{y}_\tau$
12:     **if** configuration $\in$ {RM, RM+AM} **then**
13:         $\mathbf{h}_\tau \leftarrow \rho(\mathbf{W}_r e_\tau + \mathbf{b}_r)$
14:     **else**                                       ▷ ARM / ARM+AM
15:         $\mathbf{h}_\tau \leftarrow \rho(\mathbf{W}_r e_\tau + \mathbf{W}_h \mathbf{h}_{\tau-1} + \mathbf{b}_r)$
16: **if** configuration $\in$ {RM, ARM} **then**
17:     $\mathbf{z}_t \leftarrow [\mathbf{x}_t; \mathbf{h}_{t-1}]$
18: **else**
19:     $\mathbf{z}_t \leftarrow [\mathbf{x}_t; \mathbf{h}_{t-1}; \mathbf{u}_{t-1}]$
20: $\hat{y}_t \leftarrow \mathbf{W}_o \theta(\mathbf{W}_z \mathbf{z}_t + \mathbf{b}_z) + \mathbf{b}_o$
21: **return** $\hat{y}_t$

---

**Notes.** (i) RM and RM+AM use an *instantaneous residual memory* update $\mathbf{h}_\tau \leftarrow \rho(\mathbf{W}_r e_\tau + \mathbf{b}_r)$. (ii) ARM and ARM+AM use an *accumulative residual memory* update $\mathbf{h}_\tau \leftarrow \rho(\mathbf{W}_r e_\tau + \mathbf{W}_h \mathbf{h}_{\tau-1} + \mathbf{b}_r)$. (iii) RM+AM and ARM+AM additionally carry the previous activation $\mathbf{u}_{\tau-1}$ forward as part of the state. (iv) $\theta(\cdot)$ is typically ReLU; $\rho(\cdot)$ can be ReLU or $\tanh$ depending on the ablation.

## E.2 PERFORMANCE OF VARNN VARIANTS

Table 3 reports the train and test MSE for all four VARNN configurations introduced in Section 3.4: **RM**, **RM+AM**, **ARM**, and **ARM+AM**. While the accumulative variants (**ARM**, **ARM+AM**) achieve slightly lower errors on datasets with slow drift (e.g., PM2.5), the instantaneous residual-memory models (**RM**, **RM+AM**) perform comparably across all domains with lower computational cost. For clarity, only the configuration (VARNN–RM or VARNN–RM+AM) is reported in the main results table (Table 2).

**Table 3:** Train and test mean squared error (MSE; ↓) on all datasets. *Static* models use only contemporaneous covariates $\mathbf{x}_t$; *Dynamic* models additionally incorporate lagged outputs/inputs or recurrent states. Best test scores are in **bold**.

| Model | ENERGY | | | | | | Healthcare | | | | | | Environmental | | | | | |
|---|---|---|---|---|---|---|---|---|---|---|---|---|---|---|---|---|---|---|
| | Appliances | | ETTh1 | | ETTh2 | | BIDMC HR | | BIDMC RR | | BIDMC SPO2 | | Beijing PM2.5 | | Beijing PM10 | | WEATHER | |
| | Train | Test | Train | Test | Train | Test | Train | Test | Train | Test | Train | Test | Train | Test | Train | Test | Train | Test |
| VARNN–RM | 0.00412 | 0.00328 | 0.00037 | 0.00020 | 0.00041 | 0.00048 | 0.00016 | 0.00015 | 0.00017 | 0.00031 | 0.00015 | 0.00024 | 0.00024 | 0.00026 | 0.00085 | 0.00058 | 0.00028 | 0.00045 |
| VARNN–RM+AM | 0.00378 | 0.00329 | 0.00035 | 0.00018 | 0.00044 | 0.00049 | 0.00016 | 0.00015 | 0.00017 | 0.00029 | 0.00015 | 0.00022 | 0.00022 | 0.00025 | 0.00077 | 0.00057 | 0.00037 | 0.00042 |
| VARNN–ARM | 0.00382 | 0.00330 | 0.00035 | 0.00017 | 0.00040 | 0.00046 | 0.00015 | 0.00015 | 0.00017 | 0.00029 | 0.00016 | 0.00024 | 0.00022 | 0.00025 | 0.00077 | 0.00058 | 0.00026 | 0.00031 |
| VARNN–RM+ARM | 0.00374 | 0.00329 | 0.00037 | 0.00017 | 0.00041 | 0.00051 | 0.00016 | 0.00015 | 0.00017 | 0.00029 | 0.00015 | 0.00024 | 0.00021 | 0.00024 | 0.00078 | 0.00055 | 0.00024 | 0.00032 |

As shown, the differences across variants are modest (typically within 1–2% MSE), confirming that the core residual-memory mechanism is the dominant contributor to VARNN's robustness. Therefore, the base VARNN–RM is used as the representative model in the main comparisons.

