# OpenReview forum: "Variability Aware Recursive Neural Network (VARNN): A Residual-Memory Model for Capturing Temporal Deviation in Sequence Regression Modeling"
_ICLR.cc/2026/Conference — Submitted to ICLR 2026_

### Official Review · Reviewer_3TVg · 2025-10-18

**Soundness:** 2
**Presentation:** 3
**Contribution:** 2
**Rating:** 4
**Confidence:** 3

**Summary:**

This paper introduces the Variability-Aware Recursive Neural Network (VARNN), which improves robustness in time-series regression by modeling prediction residuals as a learned “error memory.” Lightweight and efficient, VARNN updates a residual-memory pathway to recalibrate future predictions. On three experimental datasets, it showed better performance compared to the considered baselines. Variants and ablation studies further validate the design choices.

**Strengths:**

1. The paper is clearly written and easy to follow.
2. VARNN proposes a novel approach by integrating prediction residuals as a stateful memory, effectively addressing non-stationarity and distinguishing it from standard RNNs or lag-based models.
3. Its effectiveness is demonstrated through one-step-ahead forecasting with covariates on three datasets (Appliances Energy Prediction, BIDMC, Beijing), showing consistent improvements in test MSE.

**Weaknesses:**

1. The baselines considered in the paper are quite limited compared to the range of methods available in the time series forecasting/regression literature.

a) For static regression baselines, it would have been interesting to include a dynamic version using features like $X = [y_{t-w+1}, \dots, y_{t-1}, x_t] \rightarrow Y = y_t$. This could be applied to Ridge (linear), MLP, and potentially CatBoost (which tends to overfit less than Random Forest).

 b) Another way to include temporal dynamics in classical regressors is to construct features as $X = [x_t, t] \rightarrow Y = y_t$ where t is rescaled between 0 and 1 over the window w (start of window \(t=0\), end \(t=1\)). Classical regressors like Ridge, CatBoost, or the recent TabPFN could perform well with this setup.

 c) Deep learning baselines for time series, such as PatchTST, are not considered. These could potentially be adapted for one-step-ahead forecasting.

2. Only three datasets are considered, and they are relatively specific. Testing the method on a broader range of datasets, including cases with and without covariates, would strengthen the evaluation.

3. The paper focuses on one-step-ahead prediction, which is somewhat limited; extending VARNN to multi-horizon forecasting would be valuable for real-world applications.

**Questions:**

The paper is well written and interesting. However, the experimental section does not consider many elements from the literature, which makes it hard to fully trust the model’s effectiveness. Addressing (even partially) the points discussed above (in weaknesses) regarding the experiments could increase confidence in the results.

---

> ### Author Response · Authors · 2025-12-03
> **Responses to Reviewer 3TVg (part 1/3)**
>
> We appreciate your positive assessment of the clarity of the paper presentation and the novelty of our residual memory framework. Below, we address each weakness and outline the changes made in the revised version.
> >**W1: The baselines considered in the paper are quite limited compared to the range of methods available in the time series forecasting/regression literature.
> a) For static regression baselines, it would have been interesting to include a dynamic version using features like $X = [y_{t-w+1}, \dots, y_{t-1}, x_t] \rightarrow Y = y_t$. This could be applied to Ridge (linear), MLP, and potentially CatBoost (which tends to overfit less than Random Forest).
>  b) Another way to include temporal dynamics in classical regressors is to construct features as $X = [x_t, t] \rightarrow Y = y_t$ where t is rescaled between 0 and 1 over the window w (start of window (t=0), end (t=1)). Classical regressors like Ridge, CatBoost, or the recent TabPFN could perform well with this setup.
>  c) Deep learning baselines for time series, such as PatchTST, are not considered. These could potentially be adapted for one-step-ahead forecasting.**
>
>
> Thank you for your suggestion. In the revised version, we address these limitations by expanding the experiments across 13 baselines, including the SOTA transformer: PatchTST (Nie et al., 2023), and a competitive linear forecasting model: Dlinear (Zeng et al., 2023). These address strong modern models.
>
> Following your suggestion: **For static regression baselines, it would have been interesting to include a dynamic version using features like $X = [y_{t-w+1}, \dots, y_{t-1}, x_t] \rightarrow Y = y_t$. This could be applied to Ridge (linear), MLP, and potentially CatBoost (which tends to overfit less than Random Forest).**:
>
> We have added a dynamic variant of the static regressor by construction features as  $X = [y_{t-w+1}, \dots, y_{t-1}, x_t]$. We donated it as ARx-LR to distinguish it from the ARX-LR, which it follows the traditional ARX construction that uses lagged y and lagged x: $X = [y_{t-w+1}, \ldots, y_{t-1}, x_{t-w+1}, \ldots, x_t]$
>
> >**W2: Only three datasets are considered, and they are relatively specific. Testing the method on a broader range of datasets, including cases with and without covariates, would strengthen the evaluation.**
>
>
> Thank you for the suggestion and for highlighting the importance of broader empirical evaluation. In the revised version, we extended our evaluation datasets from 3 to 9 datasets across three domains: Energy, Healthcare, and Environmental, which provide a more comprehensive evaluation of the VARNN model.
>
> Importantly, we would like to emphasize that our task is single-step forecasting with exogenous covariates xt at prediction time, so the availability of covariates is essential in our task.  These selected datasets match the supervised structure of VARNN.
>
> Below is a summary of each of the 9 datasets grouped by domain. All tasks are scalar one-step forecasting with covariate xt:
>
> Note: For BIDMC HR, RR, and SPO2, the raw PPG, RESP, and ECG signals are sampled at 125 Hz. We partition each signal into non-overlapping 1-second bins and concatenate the 125 samples per channel, producing a 375-dimensional covariate vector per time step.
>
> **Table 1. Dataset summary (post-cleaning): size, missingness, feature count d, and scalar target.**
>
> | **Dataset**          | **Size**   | **Missing** | **#Feat (d)** | **Target**                               |
> |----------------------|------------|-------------|---------------|-------------------------------------------|
> | **Energy**           |            |             |               |                                           |
> | Appliances           | 19,735     | No          | 27            | Appliance energy use (Wh)                 |
> | ETTh1                | 17,420     | No          | 6             | Oil temperature                           |
> | ETTh2                | 17,420     | No          | 6             | Oil temperature                           |
> | **Healthcare (BIDMC)** |          |             |               |                                           |
> | BIDMC–HR             | 25,436     | No          | 375           | Heart rate (bpm)                          |
> | BIDMC–RR             | 25,436     | No          | 375           | Respiratory rate (breaths/min)            |
> | BIDMC–SPO2           | 25,436     | No          | 375           | Oxygen saturation (SpO2)                  |
> | **Environmental**    |            |             |               |                                           |
> | Beijing–PM2.5        | 420,768    | Yes         | 9             | PM2.5 concentration                        |
> | Beijing–PM10         | 420,768    | Yes         | 9             | PM10 concentration                         |
> | WEATHER              | 35,064     | No          | 6             | Temperature (°C)                           |
>
> More details in the revised version (Section 4)

---

> ### Author Response · Authors · 2025-12-03
> **Responses to Reviewer 3TVg (part 2/3)**
>
> >**w3*The paper focuses on one-step-ahead prediction, which is somewhat limited; extending VARNN to multi-horizon forecasting would be valuable for real-world applications.**
>
> Thank you for your valuable suggestion. We agree that multi-horizon forecasting is an important and interesting direction; however, it is a different problem setting from the one we studied in this paper. Our work is primarily framed as: one-step-ahead regression with exogenous covariates (one-step-ahead forecasting with covariates). There are many wide applications that can be adapted to real-world scenarios and are not limited to the discussed domains. For example, the VARNN can be used for job scheduling to predict I/O read or write performance in an HPC system, and so on.
> Extending VARNN to multi-step forecasting requires additional design choices (such as horizon-specific decoders or sequence-to-sequence heads) and a different experimental protocol, which are beyond the scope of the current study. We therefore keep our focus on evaluating the one-step-ahead forecasting with the covariate input setting, and briefly mention multi-horizon extensions as a promising avenue for future work.

---

> > ### Author Response · Authors · 2025-12-03
> > **Responses to Reviewer 3TVg (part 3/3)**
> >
> > For your reference, below is the result with the 13 baselines evaluated among the 9 datasets; we included the results and more discussion in the revised version (Section 5, Table 2) :
> >
> > **Train and test mean squared error (MSE; ↓) on all datasets.
> > Static models use only contemporaneous covariates x_t; Dynamic models additionally incorporate lags or in sequence. Best test scores are in bold.**
> >
> > | Model          | App Train | App Test | ETTh1 Train | ETTh1 Test | ETTh2 Train | ETTh2 Test | HR Train | HR Test | RR Train | RR Test | SPO2 Train | SPO2 Test | PM2.5 Train | PM2.5 Test | PM10 Train | PM10 Test | WEATHER Train | WEATHER Test |
> > |----------------|-----------|----------|-------------|------------|-------------|------------|----------|---------|----------|---------|------------|-----------|-------------|------------|------------|-----------|---------------|--------------|
> > | **Static (no lags)** |||||||||||||||||||
> > | LR             | 0.00799   | 0.00657  | 0.02792     | 0.03335    | 0.02243     | 0.04112    | 0.01830  | 0.02256 | 0.00984  | 0.00822 | 0.03659    | 0.03103   | 0.00212     | 0.00171    | 0.00375    | 0.00242  | 0.01516       | 0.01483     |
> > | RF             | 0.00057   | 0.04815  | 0.00234     | 0.02888    | 0.00174     | 0.03274    | 0.00031  | 0.01842 | 0.00032  | 0.00705 | 0.00057    | 0.03925   | 0.00012     | 0.00162    | 0.00025    | 0.00227  | 0.00229       | 0.01235     |
> > | MLP            | 0.00833   | 0.00778  | 0.00833     | 0.00778    | 0.02622     | 0.02049    | 0.00216  | 0.00168 | 0.01276  | 0.00933 | 0.04683    | 0.03452   | 0.00216     | 0.00168    | 0.00392    | 0.00224  | 0.01413       | 1.21301     |
> > | **Dynamic (lags)** |||||||||||||||||||
> > | DLinear        | 0.01317   | 0.01076  | 0.54476     | 0.30845    | 0.45580     | 0.43386    | 0.04563  | 0.10959 | 0.04307  | 0.08024 | 0.10516    | 0.12605   | 0.01346     | 0.01338    | 0.01477    | 0.01468  | 0.39243       | 0.40353     |
> > | ARx–LR         | 0.00395   | 0.00958  | 0.00041     | 0.04819    | 0.00030     | 0.24168    | 0.00069  | 0.03362 | 0.00041  | 0.01834 | 0.00100    | 0.05868   | 0.00028     | 0.00274    | 0.00087    | 0.00290  | 0.00042       | 0.34684     |
> > | ARX–LR         | 0.00610   | 0.00504  | 0.00067     | 0.00033    | 0.00076     | 0.00075    | 0.00023  | 0.00027 | 0.00030  | 0.00052 | 0.00028    | 0.00046   | 0.00051     | 0.00049    | 0.00144    | 0.00091  | 0.00061       | 0.00069     |
> > | NARX–RF        | 0.00079   | 0.02750  | 0.00008     | 0.00036    | 0.00008     | 0.00085    | 0.00003  | 0.00035 | 0.00004  | 0.00053 | 0.00004    | 0.00068   | 0.00005     | 0.00049    | 0.00015    | 0.00093  | 0.00006       | 0.00051     |
> > | NARX–MLP       | 0.00652   | 0.00527  | 0.00074     | 0.00036    | 0.00082     | 0.00089    | 0.00065  | 0.00079 | 0.01390  | 0.01381 | 0.00065    | 0.00098   | 0.00053     | 0.00052    | 0.00147    | 0.00094  | 0.00079       | 0.00108     |
> > | ARMAX–LR       | 0.00411   | 0.01002  | 0.00038     | 0.02952    | 0.00027     | 0.14067    | 0.00086  | 0.03092 | 0.00050  | 0.01136 | 0.00132    | 0.04648   | 0.00027     | 0.06462    | 0.00084    | 0.00429  | 0.00035       | 0.29126     |
> > | **Dynamic (sequence)** |||||||||||||||||||
> > | PatchTST       | 0.00604   | 0.00689  | 0.01560     | 0.03251    | 0.03077     | 0.04006    | 0.01961  | 0.01900 | 0.00921  | 0.00878 | 0.05707    | 0.04502   | 0.00450     | 0.00492    | 0.00673    | 0.00621  | 0.04191       | 0.03640     |
> > | RNN            | 0.00765   | 0.00697  | 0.01618     | 0.01779    | 0.01608     | 0.02246    | 0.00221  | 0.01263 | 0.00220  | 0.00811 | 0.00780    | 0.04480   | 0.00140     | 0.00150    | 0.00257    | 0.00209  | 0.00892       | 0.00973     |
> > | LSTM           | 0.00813   | 0.00652  | 0.01414     | 0.01762    | 0.01406     | 0.01997    | 0.00125  | 0.00930 | 0.00201  | 0.00746 | 0.03392    | 0.02678   | 0.00097     | 0.00143    | 0.00257    | 0.00210  | 0.00846       | 0.00928     |
> > | GRU            | 0.00824   | 0.00654  | 0.01352     | 0.02116    | 0.01576     | 0.02170    | 0.00145  | 0.00825 | 0.00274  | 0.00729 | 0.03381    | 0.02802   | 0.00110     | 0.00145    | 0.00200    | 0.00202  | 0.00834       | 0.00908     |
> > | **VARNN–RM**      | **0.00412** | **0.00328** | **0.00037**   | **0.00020**  | **0.00041**   | **0.00048**  | **0.00016** | **0.00015** | **0.00017** | **0.00031** | **0.00015**   | **0.00024** | **0.00024**   | **0.00026**  | **0.00085**   | **0.00058**  | **0.00028**     | **0.00045**   |
> > | **VARNN–RM+AM**   | **0.00378** | **0.00329** | **0.00035**   | **0.00018**  | **0.00044**   | **0.00049**  | **0.00016** | **0.00015** | **0.00017** | **0.00029** | **0.00015**   | **0.00022** | **0.00022**   | **0.00025**  | **0.00077**   | **0.00057**  | **0.00037**     | **0.00042**   |
> >
> > Overall, VARNN achieves the best test MSE across all the datasets. Specifically, modern deep forecasters such as DLinear and PatchTST are consistently less accurate under the same one-step covariate forecasting protocol.

---

### Official Review · Reviewer_58po · 2025-10-21

**Soundness:** 3
**Presentation:** 2
**Contribution:** 2
**Rating:** 4
**Confidence:** 3

**Summary:**

In this work, the author introduces VARNN, a novel recurrent architecture that explicitly models prediction residuals as a first-class memory state to improve performance in non-stationary time-series regression. The idea is motivated and empirically validated across multiple domains.

**Strengths:**

1. The idea of elevating prediction residuals to an explicit memory state is innovative and well-motivated. The paper clearly identifies a gap in how existing models (ARX, RNNs, etc.) handle temporal variability and proposes a light-weight yet effective mechanism to address it.
2. The experimental design is thorough, covering three diverse domains and multiple baseline families (static, dynamic, recurrent). The consistent superiority of VARNN across datasets strengthens the claim of robustness under non-stationarity.

**Weaknesses:**

1. There is a sentence in the Abstract whose initial letter is not capitalized. It seems not to be a complete sentence and the logical connection is also somewhat lacking. i.e., a lightweight sequence-to-one architecture...
2. While the proposed model is empirically validated, there is little theoretical analysis or intuition for why the residual memory mechanism improves generalization under non-stationarity. A deeper discussion on the connection to error-correction models or adaptive filtering would strengthen the contribution.
3. The paper does not compare with several recent strong baselines designed for non-stationary time series (e.g., N-BEATS, or transformer-based forecasters).

**Questions:**

Please see Weaknesses!

---

> ### Author Response · Authors · 2025-12-03
> **Responses to Reviewer 58po (part 1/2)**
>
> We are grateful for your comprehensive feedback! Below, we provide point-by-point responses and outline the corresponding changes made to improve our work.
>
> > **W1: While the proposed model is empirically validated, there is little theoretical analysis or intuition for why the residual memory mechanism improves generalization under non-stationarity. A deeper discussion on the connection to error-correction models or adaptive filtering would strengthen the contribution.**
>
> Thank you for your acknowledgment and for raising this important point. In the revised version, we added a new section: REPRESENTATION LEARNING OF VARIABILITY IN VARNN (Appendix B)
>
> In this section, we formally justify the intuition of the residual memory mechanism and how to track variability. We deeper discussed the key difference between VARNN and other error-based models.
>
>
>
> To clarify the distinction: VARNN encodes each one step of innovation:
>
> $$ e_{\tau} = y_{\tau} - \hat{y}_{\tau} $$
>
> into a multi-dimensional residual memory vector
>
> $$ h_{\tau} \in \mathbb{R}^{m}. $$
>
> The residual memory update used in the RM variant is:
>
> $$ h_{\tau} = \text{relu}( W_{\varepsilon} e_{\tau} + b_{\varepsilon} ). $$
>
> Or in the accumulative (ARM) variant, the innovation is composed with the previous memory state:
>
> $$ h_{\tau} = \text{relu}( W_{h} h_{\tau-1} + W_{\varepsilon} e_{\tau} + b_{\varepsilon} ). $$
>
>
> This projection allows the model to learn nonlinear, multidimensional representations of variability, such as magnitude, sign asymmetry, heteroscedasticity, burstiness, and short-term drift, which cannot be captured when the innovation is passed as a single scalar value, as in ARMAX or error injection heuristics approaches.
>
> The resulting residual memory state is then fused with the covariates in the predictive pathway via $z_{\tau} = [\ x_{\tau} \;\ h_{\tau-1} \]$, enabling each next prediction to be conditioned on a learned summary of recent deviations.
>
> Models that directly ingest the scalar residual as an input have no mechanism to structure or filter the innovation signal, which empirically leads to noise amplification and degraded performance. Under the identical training protocol of VARNN (with autoregressive and moving average orders chosen p and q=4 to match the same lag window and exogenous input X, w=5 ), across the 9 datasets, ARMAX performs consistently worse under the same conditions. For example, on ETTh1 it yields 0.02952 test MSE vs. VARNN–RM’s 0.00020, and on ETTh2 0.14067 vs. 0.00048. Similar behavior appears in healthcare and environmental datasets, where ARMAX consistently ranks among the weakest baselines ( added in Table 2).
>
> Our ablation study (Section 5.1) further shows that replacing the learned residual memory vector with a scalar residual signal greatly worsens test MSE across all domains. Using the projected residual memory reduces error by 17.5% on Appliances, 65% on PM2.5, and 98% on BIDMC–HR. These results demonstrate that VARNN learns a structured residual representation that stabilizes training, filters noise, and preserves meaningful variability patterns for next-step prediction.
>
> Thus, VARNN is fundamentally distinct from ARMAX that treats innovation as raw scalar, or ARX regressors, and standard RNN/LSTM models, which handle variability only implicitly. VARNN provides a principled, lightweight, and empirically validated approach to modeling non-stationarity via an explicit learnable residual-memory embedded dynamically.
>
>
> >**W2: There is a sentence in the Abstract whose initial letter is not capitalized. It seems not to be a complete sentence and the logical connection is also somewhat lacking. i.e., a lightweight sequence-to-one architecture...**
>
> Thanks for your notice. In the revised version, we conducted a comprehensive review and corrected grammar and phrasing issues.

---

> > ### Author Response · Authors · 2025-12-03
> > **Responses to Reviewer 58po (part 2/2)**
> >
> > >**The paper does not compare with several recent strong baselines designed for non-stationary time series (e.g., N-BEATS, or transformer-based forecasters)**
> >
> > Thank you for your valuable suggestion. In the revised versions, we expanded the experiment to include 13 baselines from three families, including the SOTA transformer, PatchTST (Nie et al., 2023), and Dlinear (Zeng et al., 2023), as well as a competitive linear forecasting model. These address strong modern models.
> > * Static: LR, RF, MLP.
> > * Dynamic (lags): ARx-LR, ARX-LR, NARX-RF, NARX-MLP, ARMAX-LR, DLinear.
> > * Dynamic (sequence): RNN, LSTM, GRU, PatchTST (Transformer)
> >
> > We evaluated the 13 baselines on nine datasets. Below are the results of dynamic models. We included the results and more discussion in the revised version (Section 5, Table 2)
> >
> > Regarding N-BEATS, we found that it is primarily designed for pure univariate autoregressive forecasting and does not incorporate multivariate covariates at each time step, which is not suitable for our problem goal.
> >
> > The updated Table shows that VARNN achieves the lowest test MSE across all datasets.
> >
> >
> > **Table 2. Train and test mean squared error (MSE; ↓) on all datasets.
> > Static models use only contemporaneous covariates x_t; Dynamic models additionally incorporate lags or recurrent states. Best test scores are in bold.**
> >
> > | Model          | App Train | App Test | ETTh1 Train | ETTh1 Test | ETTh2 Train | ETTh2 Test | HR Train | HR Test | RR Train | RR Test | SPO2 Train | SPO2 Test | PM2.5 Train | PM2.5 Test | PM10 Train | PM10 Test | WEATHER Train | WEATHER Test |
> > |----------------|-----------|----------|-------------|------------|-------------|------------|----------|---------|----------|---------|------------|-----------|-------------|------------|------------|-----------|---------------|--------------|
> > | **Dynamic (lags)** |||||||||||||||||||
> > | DLinear        | 0.01317   | 0.01076  | 0.54476     | 0.30845    | 0.45580     | 0.43386    | 0.04563  | 0.10959 | 0.04307  | 0.08024 | 0.10516    | 0.12605   | 0.01346     | 0.01338    | 0.01477    | 0.01468  | 0.39243       | 0.40353     |
> > | ARx–LR         | 0.00395   | 0.00958  | 0.00041     | 0.04819    | 0.00030     | 0.24168    | 0.00069  | 0.03362 | 0.00041  | 0.01834 | 0.00100    | 0.05868   | 0.00028     | 0.00274    | 0.00087    | 0.00290  | 0.00042       | 0.34684     |
> > | ARX–LR         | 0.00610   | 0.00504  | 0.00067     | 0.00033    | 0.00076     | 0.00075    | 0.00023  | 0.00027 | 0.00030  | 0.00052 | 0.00028    | 0.00046   | 0.00051     | 0.00049    | 0.00144    | 0.00091  | 0.00061       | 0.00069     |
> > | NARX–RF        | 0.00079   | 0.02750  | 0.00008     | 0.00036    | 0.00008     | 0.00085    | 0.00003  | 0.00035 | 0.00004  | 0.00053 | 0.00004    | 0.00068   | 0.00005     | 0.00049    | 0.00015    | 0.00093  | 0.00006       | 0.00051     |
> > | NARX–MLP       | 0.00652   | 0.00527  | 0.00074     | 0.00036    | 0.00082     | 0.00089    | 0.00065  | 0.00079 | 0.01390  | 0.01381 | 0.00065    | 0.00098   | 0.00053     | 0.00052    | 0.00147    | 0.00094  | 0.00079       | 0.00108     |
> > | ARMAX–LR       | 0.00411   | 0.01002  | 0.00038     | 0.02952    | 0.00027     | 0.14067    | 0.00086  | 0.03092 | 0.00050  | 0.01136 | 0.00132    | 0.04648   | 0.00027     | 0.06462    | 0.00084    | 0.00429  | 0.00035       | 0.29126     |
> > | **Dynamic (sequence)** |||||||||||||||||||
> > | PatchTST       | 0.00604   | 0.00689  | 0.01560     | 0.03251    | 0.03077     | 0.04006    | 0.01961  | 0.01900 | 0.00921  | 0.00878 | 0.05707    | 0.04502   | 0.00450     | 0.00492    | 0.00673    | 0.00621  | 0.04191       | 0.03640     |
> > | RNN            | 0.00765   | 0.00697  | 0.01618     | 0.01779    | 0.01608     | 0.02246    | 0.00221  | 0.01263 | 0.00220  | 0.00811 | 0.00780    | 0.04480   | 0.00140     | 0.00150    | 0.00257    | 0.00209  | 0.00892       | 0.00973     |
> > | LSTM           | 0.00813   | 0.00652  | 0.01414     | 0.01762    | 0.01406     | 0.01997    | 0.00125  | 0.00930 | 0.00201  | 0.00746 | 0.03392    | 0.02678   | 0.00097     | 0.00143    | 0.00257    | 0.00210  | 0.00846       | 0.00928     |
> > | GRU            | 0.00824   | 0.00654  | 0.01352     | 0.02116    | 0.01576     | 0.02170    | 0.00145  | 0.00825 | 0.00274  | 0.00729 | 0.03381    | 0.02802   | 0.00110     | 0.00145    | 0.00200    | 0.00202  | 0.00834       | 0.00908     |
> > | **VARNN–RM**      | **0.00412** | **0.00328** | **0.00037**   | **0.00020**  | **0.00041**   | **0.00048**  | **0.00016** | **0.00015** | **0.00017** | **0.00031** | **0.00015**   | **0.00024** | **0.00024**   | **0.00026**  | **0.00085**   | **0.00058**  | **0.00028**     | **0.00045**   |
> > | **VARNN–RM+AM**   | **0.00378** | **0.00329** | **0.00035**   | **0.00018**  | **0.00044**   | **0.00049**  | **0.00016** | **0.00015** | **0.00017** | **0.00029** | **0.00015**   | **0.00022** | **0.00022**   | **0.00025**  | **0.00077**   | **0.00057**  | **0.00037**     | **0.00042**   |

---

### Official Review · Reviewer_gohe · 2025-10-31

**Soundness:** 2
**Presentation:** 2
**Contribution:** 2
**Rating:** 2
**Confidence:** 3

**Summary:**

The paper proposes the Variability-Aware Recursive Neural Network (VARNN) — a residual-memory architecture for time-series regression. The key idea is to explicitly model prediction residuals as a recurrent state to handle non-stationarity, variability, and drift in temporal data. The authors evaluate several variants (instantaneous vs. accumulative residual memory, with/without activation carry-over) on three datasets (energy, healthcare, environmental) and report consistent MSE improvements over ARX/NARX and RNN baselines.

**Strengths:**

1. The idea of using residuals as a first-class state is conceptually simple and potentially useful.
2. Empirical evaluation spans multiple domains and baselines, with consistent performance gains reported.

**Weaknesses:**

1. The paper is very difficult to follow: grammar and phrasing issues are pervasive from the first paragraph (“Regression is is one of the fundamental tasks/ a fundamental task…”). The exposition is verbose, repetitive, and at times inconsistent in notation. There are many typos and formatting errors that hinder readability.
2. The paper claims to propose a new neural architecture, but offers no theoretical analysis or formal justification.
3. Comparisons are relatively weak. Only simple baselines (ARX, MLP, RNN) are included. Missing comparisons with modern strong regressors for time series, including Transformers (Informer, TFT, PatchTST) or State-space models (SSM/Mamba), which are natural choices for regression under non-stationarity.
4. The proposed mechanism, feeding residuals back as input, is not conceptually new.

**Questions:**

see above

---

> ### Author Response · Authors · 2025-12-03
> **Responses to Reviewer gohe (part 1/2)**
>
> We appreciate the comprehensive feedback and the highlighting of several aspects! Below, we provide point-by-point responses and outline the corresponding changes made to improve our work.
>
> > **W1: The paper is very difficult to follow: grammar and phrasing issues are pervasive from the first paragraph (“Regression is is one of the fundamental tasks/ a fundamental task…”). The exposition is verbose, repetitive, and at times inconsistent in notation. There are many typos and formatting errors that hinder readability.**
>
> We appreciate this observation. In the revised version:
> * We conducted a comprehensive review and corrected grammar and phrasing issues.
> * We added a modular presentation (Predictor block, Residual memory block)
> * We fix the formal problem setup, window notation, and model equations with consistent notations.
>
> > **W2: The proposed mechanism, feeding residuals back as input, is not conceptually new.**
>
> Thank you for raising this point. We agree that innovation-based mechanisms exist in classical state-space models and in ARMAX models. Our claim is not that innovation feedback is new, but that VARNN is the first neural architecture that explicitly treats the innovation (residual) as the primary recurrent state via a learned vector embedding, rather than using the residual as a raw scalar or absorbing variability implicitly through hidden-state dynamics.
>
> To clarify the distinction: VARNN encodes each one step of innovation:
>
> $$ e_{\tau} = y_{\tau} - \hat{y}_{\tau} $$
>
> into a multidimensional residual memory vector
>
> $$ h_{\tau} \in \mathbb{R}^{m}. $$
>
> The residual memory update used in the RM variant is:
>
> $$ h_{\tau} = \text{relu}( W_{\varepsilon} e_{\tau} + b_{\varepsilon} ). $$
>
> Or in the accumulative (ARM) variant, the innovation is composed with the previous memory state:
>
> $$ h_{\tau} = \text{relu}( W_{h} h_{\tau-1} + W_{\varepsilon} e_{\tau} + b_{\varepsilon} ). $$
>
>
> This projection allows the model to learn nonlinear, multidimensional representations of variability, such as magnitude, sign asymmetry, heteroscedasticity, burstiness, and short-term drift, which cannot be captured when the innovation is passed as a single scalar value, as in ARMAX or error injection heuristics approaches.
>
> The resulting residual memory state is then fused with the covariates in the predictive pathway via $z_{\tau} = [\ x_{\tau} \;\ h_{\tau-1} \]$, enabling each next prediction to be conditioned on a learned summary of recent deviations.
>
> Models that directly ingest the scalar residual as an input have no mechanism to structure or filter the innovation signal, which empirically leads to noise amplification and degraded performance. Under the identical training protocol of VARNN (with autoregressive and moving average orders chosen p and q=4 to match the same lag window and exogenous input X, w=5 ), across the nine datasets, ARMAX performs consistently worse under the same conditions. For example, on ETTh1 it yields 0.02952 test MSE vs. VARNN–RM’s 0.00020, and on ETTh2 0.14067 vs. 0.00048. Similar behavior appears in healthcare and environmental datasets, where ARMAX consistently ranks among the weakest baselines ( added in Table 2).
>
> Our ablation study (Section 5.1) further shows that replacing the learned residual memory vector with a scalar residual signal considerably worsens test MSE across all domains. Using the projected residual memory reduces error by 17.5% on Appliances, 65% on PM2.5, and 98% on BIDMC–HR.
>
> Thus, VARNN is fundamentally distinct from ARMAX, which treats innovation as raw scalar, or ARX regressors, and standard RNN/LSTM models, which handle variability only implicitly. VARNN provides a principled, lightweight, and empirically validated approach to modeling non-stationarity via an explicit learnable residual-memory embedded dynamically.
>
> Correspondingly, we have added this clarification in Appendix B of the revised version.
>
> > **W3: The paper claims to propose a new neural architecture, but offers no theoretical analysis or formal justification.**
>
> Thanks for your feedback. we have included in the current revised version two analyses and a formal justification:
>
> * REPRESENTATION LEARNING OF VARIABILITY IN VARNN (Appendix B):
> We explain, as mentioned above, why and how Varnn learn and tracks variability using RM and ARM residual memory projection
> * VARNN VS STANDARD RECURRENT MODELS ( Section 3.7):
> We justify the key differences between the standard recurrent model and the VARNN objective and recurrent state update, where the RNN hidden state stands for the prior prediction state. In contrast, VARNN's hidden state is the residual projection of the most recent step (RM variant) or the accumulation of all window steps (as in the ARM variant), which makes VARNN use innovation as a first-class signal that conditions the step, making it stable in non-stationary environments.

---

> ### Author Response · Authors · 2025-12-03
> **Responses to Reviewer gohe (part 2/2)**
>
> > **Comparisons are relatively weak. Only simple baselines (ARX, MLP, RNN) are included. Missing comparisons with modern strong regressors for time series, including Transformers (Informer, TFT, PatchTST) or State-space models (SSM/Mamba), which are natural choices for regression under non-stationarity.**
>
> Thank you for your valuable suggestion. In the revised versions, we expanded the experiment to include 13 baselines from three families, including the SOTA transformer, PatchTST (Nie et al., 2023), and Dlinear (Zeng et al., 2023), as well as a competitive linear forecasting model. These address strong modern models.
> * Static: LR, RF, MLP.
> * Dynamic (lags): ARx-LR, ARX-LR, NARX-RF, NARX-MLP, ARMAX-LR, DLinear.
> * Dynamic (sequence): RNN, LSTM, GRU, PatchTST (Transformer)
>
> We evaluated the 13 baselines on nine datasets. Below are the results of dynamic models. We included the results and more discussion in the revised version (Section 5, Table 2)
>
> The updated Table shows that VARNN achieves the lowest test MSE across all datasets, demonstrating the value of an explicit residual vector representation in non-stationary environments.
>
>
> **Table 2. Train and test mean squared error (MSE; ↓) on all datasets.
> Static models use only contemporaneous covariates x_t; Dynamic models additionally incorporate lags or recurrent states. Best test scores are in bold.**
>
> | Model          | App Train | App Test | ETTh1 Train | ETTh1 Test | ETTh2 Train | ETTh2 Test | HR Train | HR Test | RR Train | RR Test | SPO2 Train | SPO2 Test | PM2.5 Train | PM2.5 Test | PM10 Train | PM10 Test | WEATHER Train | WEATHER Test |
> |----------------|-----------|----------|-------------|------------|-------------|------------|----------|---------|----------|---------|------------|-----------|-------------|------------|------------|-----------|---------------|--------------|
> | **Dynamic (lags)** |||||||||||||||||||
> | DLinear        | 0.01317   | 0.01076  | 0.54476     | 0.30845    | 0.45580     | 0.43386    | 0.04563  | 0.10959 | 0.04307  | 0.08024 | 0.10516    | 0.12605   | 0.01346     | 0.01338    | 0.01477    | 0.01468  | 0.39243       | 0.40353     |
> | ARx–LR         | 0.00395   | 0.00958  | 0.00041     | 0.04819    | 0.00030     | 0.24168    | 0.00069  | 0.03362 | 0.00041  | 0.01834 | 0.00100    | 0.05868   | 0.00028     | 0.00274    | 0.00087    | 0.00290  | 0.00042       | 0.34684     |
> | ARX–LR         | 0.00610   | 0.00504  | 0.00067     | 0.00033    | 0.00076     | 0.00075    | 0.00023  | 0.00027 | 0.00030  | 0.00052 | 0.00028    | 0.00046   | 0.00051     | 0.00049    | 0.00144    | 0.00091  | 0.00061       | 0.00069     |
> | NARX–RF        | 0.00079   | 0.02750  | 0.00008     | 0.00036    | 0.00008     | 0.00085    | 0.00003  | 0.00035 | 0.00004  | 0.00053 | 0.00004    | 0.00068   | 0.00005     | 0.00049    | 0.00015    | 0.00093  | 0.00006       | 0.00051     |
> | NARX–MLP       | 0.00652   | 0.00527  | 0.00074     | 0.00036    | 0.00082     | 0.00089    | 0.00065  | 0.00079 | 0.01390  | 0.01381 | 0.00065    | 0.00098   | 0.00053     | 0.00052    | 0.00147    | 0.00094  | 0.00079       | 0.00108     |
> | ARMAX–LR       | 0.00411   | 0.01002  | 0.00038     | 0.02952    | 0.00027     | 0.14067    | 0.00086  | 0.03092 | 0.00050  | 0.01136 | 0.00132    | 0.04648   | 0.00027     | 0.06462    | 0.00084    | 0.00429  | 0.00035       | 0.29126     |
> | **Dynamic (sequence)** |||||||||||||||||||
> | PatchTST       | 0.00604   | 0.00689  | 0.01560     | 0.03251    | 0.03077     | 0.04006    | 0.01961  | 0.01900 | 0.00921  | 0.00878 | 0.05707    | 0.04502   | 0.00450     | 0.00492    | 0.00673    | 0.00621  | 0.04191       | 0.03640     |
> | RNN            | 0.00765   | 0.00697  | 0.01618     | 0.01779    | 0.01608     | 0.02246    | 0.00221  | 0.01263 | 0.00220  | 0.00811 | 0.00780    | 0.04480   | 0.00140     | 0.00150    | 0.00257    | 0.00209  | 0.00892       | 0.00973     |
> | LSTM           | 0.00813   | 0.00652  | 0.01414     | 0.01762    | 0.01406     | 0.01997    | 0.00125  | 0.00930 | 0.00201  | 0.00746 | 0.03392    | 0.02678   | 0.00097     | 0.00143    | 0.00257    | 0.00210  | 0.00846       | 0.00928     |
> | GRU            | 0.00824   | 0.00654  | 0.01352     | 0.02116    | 0.01576     | 0.02170    | 0.00145  | 0.00825 | 0.00274  | 0.00729 | 0.03381    | 0.02802   | 0.00110     | 0.00145    | 0.00200    | 0.00202  | 0.00834       | 0.00908     |
> | **VARNN–RM**      | **0.00412** | **0.00328** | **0.00037**   | **0.00020**  | **0.00041**   | **0.00048**  | **0.00016** | **0.00015** | **0.00017** | **0.00031** | **0.00015**   | **0.00024** | **0.00024**   | **0.00026**  | **0.00085**   | **0.00058**  | **0.00028**     | **0.00045**   |
> | **VARNN–RM+AM**   | **0.00378** | **0.00329** | **0.00035**   | **0.00018**  | **0.00044**   | **0.00049**  | **0.00016** | **0.00015** | **0.00017** | **0.00029** | **0.00015**   | **0.00022** | **0.00022**   | **0.00025**  | **0.00077**   | **0.00057**  | **0.00037**     | **0.00042**   |

---

### Official Review · Reviewer_yCb6 · 2025-11-07

**Soundness:** 2
**Presentation:** 3
**Contribution:** 1
**Rating:** 2
**Confidence:** 4

**Summary:**

This paper introduces a new neural architecture, VARNN, a neural for time-series "regression" that explicitly incorporates prediction errors (called innovations) as a recurrent state. Unlike standard RNNs that implicitly absorb variability into hidden states, VARNN maintains a dedicated "residual memory" that tracks recent prediction errors and uses this signal to calibrate subsequent predictions. Experiments are conducted to  evaluate the proposed approach across three datasets (appliance energy, healthcare heart rate, and air quality), compared to a few baselines.

**Strengths:**

* The core idea of the paper is well motivated and well explained.
*  The architecture is clearly described, flexible and lightweight.
* The paper presents ablation experiments and several variants to  apprehend  the proposed architecture.

**Weaknesses:**

* While the paper frames the problem as "time-series regression," the task is fundamentally one-step-ahead forecasting. This framing issue leads to a critical gap in the literature review: none of the recent state-of-the-art forecasting methods are cited or discussed (including PatchTST (Nie et al., 2023), Autoformer (Wu et al., 2021), DLinear (Zeng et al., 2023), etc). This omission is problematic because these methods represent the current competitive landscape. Without acknowledging or comparing against them, it is impossible to assess VARNN's true contribution to the forecasting literature.

* The core claim—that "elevating innovations  to first-class signals" is novel—overlooks substantial prior work as  Kalman Filters  and litterature in deep learning related to state space models, or ARMAX models.

* The experimental evaluation is too narrow to support the paper's claims: Only vanilla RNN is evaluated among recurrent models; LSTM and GRU are absent, despite being standard baselines. No comparison with methods designed for non-stationarity: The related work cites RevIN (Kim et al., 2022), GARCH+DL (Han et al., 2024), and de-stationing methods, but none are evaluated. No classical forecasting baselines: ARIMA, Exponential Smoothing. No modern deep forecasting methods: DLinear,  or any of the Transformer-based methods mentioned above. The current baselines function as a proof-of-concept that VARNN's architecture is functional, but they do not establish its value relative to the state-of-the-art.

* Only three datasets are evaluated, all relatively small and domain-specific. No evaluation on standard forecasting benchmarks (M4, Electricity, ETT, Traffic, Weather) that would enable comparison with published results

**Questions:**

See above.

---

> ### Author Response · Authors · 2025-12-03
> **Responses to Reviewer yCb6 (part 1/4)**
>
> We are grateful for your comprehensive feedback! Below, we provide point-by-point responses and outline the corresponding changes made to improve our work.
>
> > **W1: The core claim—that "elevating innovations to first-class signals" is novel—overlooks substantial prior work such as Kalman Filters, state-space models, or ARMAX models.**
>
> Thank you for raising this point. We agree that innovation-based mechanisms exist in classical state-space models and ARMAX. Our claim is not that innovation feedback is new, but that VARNN is the first neural architecture that explicitly treats the innovation (residual) as the primary recurrent state via a learned vector embedding, rather than using the residual as a raw scalar or absorbing variability implicitly through hidden-state dynamics.
>
> To clarify the distinction: VARNN encodes each one step of innovation:
>
> $$ e_{\tau} = y_{\tau} - \hat{y}_{\tau} $$
>
> into a multi-dimensional residual memory vector
>
> $$ h_{\tau} \in \mathbb{R}^{m}. $$
>
> The residual memory update used in the RM variant is:
>
> $$ h_{\tau} = \text{relu}( W_{\varepsilon} e_{\tau} + b_{\varepsilon} ). $$
>
> Or in the accumulative (ARM) variant, the innovation is composed with the previous memory state:
>
> $$ h_{\tau} = \text{relu}( W_{h} h_{\tau-1} + W_{\varepsilon} e_{\tau} + b_{\varepsilon} ). $$
>
>
> This projection allows the model to learn nonlinear, multidimensional representations of variability, such as magnitude, sign asymmetry, heteroscedasticity, burstiness, and short-term drift, which cannot be captured when the innovation is passed as a single scalar value, as in ARMAX or error injection heuristics approaches.
>
> The resulting residual memory state is then fused with the covariates in the predictive pathway via $z_{\tau} = [\ x_{\tau} \;\ h_{\tau-1} \]$, enabling each next prediction to be conditioned on a learned summary of recent deviations.
>
> Models that directly ingest the scalar residual as an input have no mechanism to structure or filter the innovation signal, which empirically leads to noise amplification and degraded performance. Under the identical training protocol of VARNN (with autoregressive and moving average orders chosen p and q=4 to match the same lag window and exogenous input X, w=5 ), across the 9 datasets, ARMAX performs consistently worse under the same conditions. For example, on ETTh1 it yields 0.02952 test MSE vs. VARNN–RM’s 0.00020, and on ETTh2 0.14067 vs. 0.00048. Similar behavior appears in healthcare and environmental datasets, where ARMAX consistently ranks among the weakest baselines ( added in Table 2).
>
> Our ablation study (Section 5.1) further shows that replacing the learned residual memory vector with a scalar residual signal greatly worsens test MSE across all domains. Using the projected residual memory reduces error by 17.5% on Appliances, 65% on PM2.5, and 98% on BIDMC–HR. These results demonstrate that VARNN learns a structured residual representation (not merely reuse prediction error) that stabilizes training, filters noise, and preserves meaningful variability patterns for next-step prediction.
>
> Thus, VARNN is fundamentally distinct from ARMAX that treats innovation as raw scalar, or ARX regressors, and standard RNN/LSTM models, which handle variability only implicitly. VARNN provides a principled, lightweight, and empirically validated approach to modeling non-stationarity via an explicit learnable residual-memory embedded dynamically.
>
> Correspondingly, we have added this clarification in Appendix B of the revised version.

---

> ### Author Response · Authors · 2025-12-03
> **Responses to Reviewer yCb6 (part 2/4)**
>
> > **W2: While the paper frames the problem as "time-series regression," the task is fundamentally one-step-ahead forecasting. This framing issue leads to a critical gap in the literature review: none of the recent state-of-the-art forecasting methods are cited or discussed (including PatchTST (Nie et al., 2023), Autoformer (Wu et al., 2021), DLinear (Zeng et al., 2023), etc). This omission is problematic because these methods represent the current competitive landscape. Without acknowledging or comparing against them, it is impossible to assess VARNN's true contribution to the forecasting literature.**
>
> >**W3: The experimental evaluation is too narrow to support the paper's claims: Only vanilla RNN is evaluated among recurrent models; LSTM and GRU are absent, despite being standard baselines. No comparison with methods designed for non-stationarity: The related work cites RevIN (Kim et al., 2022), GARCH+DL (Han et al., 2024), and de-stationing methods, but none are evaluated. No classical forecasting baselines: ARIMA, Exponential Smoothing. No modern deep forecasting methods: DLinear, or any of the Transformer-based methods mentioned above. The current baselines function as a proof-of-concept that VARNN's architecture is functional, but they do not establish its value relative to the state-of-the-art.**
>
> We really appreciate your acknowledgment that our approach, VARNN, is a functional architecture, and thank you for the suggestions you mentioned to help strengthen the paper's validation and get it accepted. Below are the key suggestions and the exchange that we made in the revised version:
>
> **Clarification of the Task Framing:**
>
> * Thank you for raising this point, and we apologize for the confusion. In the original version, we mentioned it in the introduction section,  second paragraph, that:  This motivates time series regression, where the objective is to predict the output at each timestep using the current input and a short history of recent information **(one-step-ahead, per-timestep supervised prediction)** Hyndman & Athanasopoulos (2021); Brockwell & Davis (2002).
>
> * Following your notable feedback, we explicitly described the task as **one-step-ahead forecasting** in the introduction section (section 1) as well as in the experiment setting (section 4).
>
> **Addressing the literature gap and including the SOTA Forecasting models:**
> * We appreciate your pointing out. While in the earlier version we briefly referenced and acknowledged the transformer-based approach (e.g., Infomer), we agree that a more detailed discussion and a direct comparison were necessary to validate our proposed model.
>
> * Accordingly, we have currently added in the revised version both the State-of-the-art Transformer forecasters PatchTST (Nie et al., 2023) and Dlinear (Zeng et al., 2023) in the related work and as baselines in the experiments (Sec. 4; and Table 2). These represent the current competitive landscape in forecasting, and we ensured identical preprocessing and training conditions. These comparisons demonstrate that VARNN consistently outperforms these SOTA forecasting models across the same window length and under the one-step-ahead forecasting protocol with covariate inputs.
>
> * We also included the following baselines: LSTM, GRU, ARMAX. Thus, the current revised version of our paper compared VARNN vs 13 baselines in total.
>
> **Regrading the absence of evaluating RevIN, GARCH+DL:**
>
> We appreciate this suggestion. We would like to mention that RevIN is indeed widely used as a technique of a normalization module, not a forecasting architecture, which can be applied to any model (including VARNN, LSTM, PatchTST). In this paper, our goal is to make an architectural contribution rather than to propose a normalization strategy. For the GARCH family, it is a volatility model indicator that estimates whether volatility is high or low, which is used to analyze time series data in finance, so it is not aligned with our regression task with exogenous covariates. We mentioned them in the related work as approaches that were used in a non-stationary environment.
>
>
> **Regarding the absence of classical forecasting baselines (ex, ARIMA)**
>
> Our problem is formally defined as multivariate exogenous covariate inputs with known xt, and one-step ahead forecasting for yt, so ARIMA and standard forecasting commonly cannot incorporate covariates. Instead, the closest classical baselines that support exogenous inputs are: ARX, NARX, and ARMAX. In the revised version, we extend our experiments and include all these baselines, as these models already serve as generalized approaches to classical forecasting baselines based on multivariate regression with noise smoothing.
>
> In the following comment, we will present the detailed results table for the added datasets.

---

> > ### Author Response · Authors · 2025-12-03
> > **Responses to Reviewer yCb6 (part 3/4)**
> >
> > > **W4: Only three datasets are evaluated, all relatively small and domain-specific. No evaluation on standard forecasting benchmarks (M4, Electricity, ETT, Traffic, Weather) that would enable comparison with published results**
> >
> > Thank you for the suggestion and for highlighting the importance of broader empirical evaluation. In the revised version, we extended our evaluation datasets from 3 to 9 datasets across three domains: Energy, Healthcare, and Environmental, which provide a more comprehensive evaluation of the VARNN model.
> >
> > Importantly, we would like to note that our task is single-step forecasting with exogenous covariates at prediction time, so many of the suggested standard long-horizon benchmarks (such as M4, Traffic, ECL) do not provide multivariate covariate-target pairs (for instance, the Traffic dataset consists of only  862 independent sensors, and there is no single dependent variable that can be used as (y)) making them incompatible for our specific modeling setup. Thus, we selected the suggested datasets that match the supervised structure of VARNN.
> >
> > Below are the details of each of the datasets benchmarked:
> >
> > We use nine popular datasets grouped by domain (summarized in Table 1). All tasks are scalar one-step forecasting.
> >
> > - **Energy.**
> >   (i) **Appliances Energy Prediction** (Appliances): household energy use with indoor/outdoor weather and occupancy covariates; target is appliance energy consumption (Wh).
> >   (ii) **ETTh1** and (iii) **ETTh2**: electricity transformer temperature benchmarks with hourly load and meteorological features; the target is the oil temperature using exogenous covariates.
> >
> > - **Healthcare.**
> >   We use three scalar targets derived from the BIDMC PPG/ECG recordings:
> >   (iv) **BIDMC HR** (heart rate),
> >   (v) **BIDMC RR** (respiratory rate),
> >   (vi) **BIDMC SPO2** (oxygen saturation).
> >   All tasks use waveform-derived covariates constructed from PPG, RESP, and ECG channels.
> >
> > - **Environmental.**
> >   (vii) **Beijing PM2.5** and (viii) **Beijing PM10**: air-quality and meteorological variables from 12 monitoring stations; targets are PM2.5 and PM10 concentrations respectively.
> >   (ix) **WEATHER**: multivariate meteorological time series with temperature as the scalar target.
> >
> > Note: For BIDMC HR, RR, and SPO2, the raw PPG, RESP, and ECG signals are sampled at 125 Hz. We partition each signal into non-overlapping 1-second bins and concatenate the 125 samples per channel, producing a 375-dimensional covariate vector per time step.
> >
> > I have attached Table 1 below:
> >
> > **Table 1. Dataset summary (post-cleaning): size, missingness, feature count d, and scalar target.**
> >
> > | **Dataset**          | **Size**   | **Missing** | **#Feat (d)** | **Target**                               |
> > |----------------------|------------|-------------|---------------|-------------------------------------------|
> > | **Energy**           |            |             |               |                                           |
> > | Appliances           | 19,735     | No          | 27            | Appliance energy use (Wh)                 |
> > | ETTh1                | 17,420     | No          | 6             | Oil temperature                           |
> > | ETTh2                | 17,420     | No          | 6             | Oil temperature                           |
> > | **Healthcare (BIDMC)** |          |             |               |                                           |
> > | BIDMC–HR             | 25,436     | No          | 375           | Heart rate (bpm)                          |
> > | BIDMC–RR             | 25,436     | No          | 375           | Respiratory rate (breaths/min)            |
> > | BIDMC–SPO2           | 25,436     | No          | 375           | Oxygen saturation (SpO2)                  |
> > | **Environmental**    |            |             |               |                                           |
> > | Beijing–PM2.5        | 420,768    | Yes         | 9             | PM2.5 concentration                        |
> > | Beijing–PM10         | 420,768    | Yes         | 9             | PM10 concentration                         |
> > | WEATHER              | 35,064     | No          | 6             | Temperature (°C)                           |

---

> > > ### Author Response · Authors · 2025-12-03
> > > **Responses to Reviewer yCb6 (part 4/4)**
> > >
> > > Below is the result with the 13 baselines evaluated among the 9 datasets; we included the result and more discussion in the revised version (Section 5, Table 2) :
> > >
> > > **Table 2. Train and test mean squared error (MSE; ↓) on all datasets.
> > > Static models use only contemporaneous covariates x_t; Dynamic models additionally incorporate lags or recurrent states. Best test scores are in bold.**
> > >
> > > | Model          | App Train | App Test | ETTh1 Train | ETTh1 Test | ETTh2 Train | ETTh2 Test | HR Train | HR Test | RR Train | RR Test | SPO2 Train | SPO2 Test | PM2.5 Train | PM2.5 Test | PM10 Train | PM10 Test | WEATHER Train | WEATHER Test |
> > > |----------------|-----------|----------|-------------|------------|-------------|------------|----------|---------|----------|---------|------------|-----------|-------------|------------|------------|-----------|---------------|--------------|
> > > | **Static (no lags)** |||||||||||||||||||
> > > | LR             | 0.00799   | 0.00657  | 0.02792     | 0.03335    | 0.02243     | 0.04112    | 0.01830  | 0.02256 | 0.00984  | 0.00822 | 0.03659    | 0.03103   | 0.00212     | 0.00171    | 0.00375    | 0.00242  | 0.01516       | 0.01483     |
> > > | RF             | 0.00057   | 0.04815  | 0.00234     | 0.02888    | 0.00174     | 0.03274    | 0.00031  | 0.01842 | 0.00032  | 0.00705 | 0.00057    | 0.03925   | 0.00012     | 0.00162    | 0.00025    | 0.00227  | 0.00229       | 0.01235     |
> > > | MLP            | 0.00833   | 0.00778  | 0.00833     | 0.00778    | 0.02622     | 0.02049    | 0.00216  | 0.00168 | 0.01276  | 0.00933 | 0.04683    | 0.03452   | 0.00216     | 0.00168    | 0.00392    | 0.00224  | 0.01413       | 1.21301     |
> > > | **Dynamic (lags)** |||||||||||||||||||
> > > | DLinear        | 0.01317   | 0.01076  | 0.54476     | 0.30845    | 0.45580     | 0.43386    | 0.04563  | 0.10959 | 0.04307  | 0.08024 | 0.10516    | 0.12605   | 0.01346     | 0.01338    | 0.01477    | 0.01468  | 0.39243       | 0.40353     |
> > > | ARx–LR         | 0.00395   | 0.00958  | 0.00041     | 0.04819    | 0.00030     | 0.24168    | 0.00069  | 0.03362 | 0.00041  | 0.01834 | 0.00100    | 0.05868   | 0.00028     | 0.00274    | 0.00087    | 0.00290  | 0.00042       | 0.34684     |
> > > | ARX–LR         | 0.00610   | 0.00504  | 0.00067     | 0.00033    | 0.00076     | 0.00075    | 0.00023  | 0.00027 | 0.00030  | 0.00052 | 0.00028    | 0.00046   | 0.00051     | 0.00049    | 0.00144    | 0.00091  | 0.00061       | 0.00069     |
> > > | NARX–RF        | 0.00079   | 0.02750  | 0.00008     | 0.00036    | 0.00008     | 0.00085    | 0.00003  | 0.00035 | 0.00004  | 0.00053 | 0.00004    | 0.00068   | 0.00005     | 0.00049    | 0.00015    | 0.00093  | 0.00006       | 0.00051     |
> > > | NARX–MLP       | 0.00652   | 0.00527  | 0.00074     | 0.00036    | 0.00082     | 0.00089    | 0.00065  | 0.00079 | 0.01390  | 0.01381 | 0.00065    | 0.00098   | 0.00053     | 0.00052    | 0.00147    | 0.00094  | 0.00079       | 0.00108     |
> > > | ARMAX–LR       | 0.00411   | 0.01002  | 0.00038     | 0.02952    | 0.00027     | 0.14067    | 0.00086  | 0.03092 | 0.00050  | 0.01136 | 0.00132    | 0.04648   | 0.00027     | 0.06462    | 0.00084    | 0.00429  | 0.00035       | 0.29126     |
> > > | **Dynamic (sequence)** |||||||||||||||||||
> > > | PatchTST       | 0.00604   | 0.00689  | 0.01560     | 0.03251    | 0.03077     | 0.04006    | 0.01961  | 0.01900 | 0.00921  | 0.00878 | 0.05707    | 0.04502   | 0.00450     | 0.00492    | 0.00673    | 0.00621  | 0.04191       | 0.03640     |
> > > | RNN            | 0.00765   | 0.00697  | 0.01618     | 0.01779    | 0.01608     | 0.02246    | 0.00221  | 0.01263 | 0.00220  | 0.00811 | 0.00780    | 0.04480   | 0.00140     | 0.00150    | 0.00257    | 0.00209  | 0.00892       | 0.00973     |
> > > | LSTM           | 0.00813   | 0.00652  | 0.01414     | 0.01762    | 0.01406     | 0.01997    | 0.00125  | 0.00930 | 0.00201  | 0.00746 | 0.03392    | 0.02678   | 0.00097     | 0.00143    | 0.00257    | 0.00210  | 0.00846       | 0.00928     |
> > > | GRU            | 0.00824   | 0.00654  | 0.01352     | 0.02116    | 0.01576     | 0.02170    | 0.00145  | 0.00825 | 0.00274  | 0.00729 | 0.03381    | 0.02802   | 0.00110     | 0.00145    | 0.00200    | 0.00202  | 0.00834       | 0.00908     |
> > > | **VARNN–RM**      | **0.00412** | **0.00328** | **0.00037**   | **0.00020**  | **0.00041**   | **0.00048**  | **0.00016** | **0.00015** | **0.00017** | **0.00031** | **0.00015**   | **0.00024** | **0.00024**   | **0.00026**  | **0.00085**   | **0.00058**  | **0.00028**     | **0.00045**   |
> > > | **VARNN–RM+AM**   | **0.00378** | **0.00329** | **0.00035**   | **0.00018**  | **0.00044**   | **0.00049**  | **0.00016** | **0.00015** | **0.00017** | **0.00029** | **0.00015**   | **0.00022** | **0.00022**   | **0.00025**  | **0.00077**   | **0.00057**  | **0.00037**     | **0.00042**   |
> > >
> > > Overall, VARNN achieves the best test MSE across all the datasets. Specifically, modern deep forecasters  such as DLinear and PatchTST are consistently less accurate under the same one-step covariate forecasting protocol.

---

### Meta-Review · Area_Chair_QEzf · 2025-12-06

**Summary:**

The paper proposes VARNN, a residual-memory RNN for one-step-ahead forecasting. The rebuttal adds more datasets and baselines (including PatchTST and DLinear). The work is empirically solid in this narrow one-step setting, but novelty and scope (no commonly used multi-horizon evaluation, no theory beyond intuition) remain limited.  The comparison with PatchTST comparison relies on a non-standard, easier protocol.

**Reviewer Concerns:**

yCb6: Partially address empirical results concern and the problem setup.

gohe: Partially address empirical results concern but miss theoretical justification.

58po: Partially address empirical results concern but miss theoretical justification.

3TVg: Partially address empirical results concern and novelty / problem setup.

**Reviewer Scores:**

All reviews may raise their score by 1 as the authors partially address the empirical result concern.

---

### Decision · Program_Chairs · 2026-01-26

Reject